# BehaviorDEPOT is a simple, flexible tool for automated behavioral detection based on markerless pose tracking

Christopher J Gabriel[1,2], Zachary Zeidler[1], Benita Jin[1,3], Changliang Guo[4], Caitlin M Goodpaster[2], Adrienne Q Kashay[5], Anna Wu[1], Molly Delaney[5], Jovian Cheung[5], Lauren E DiFazio[6], Melissa J Sharpe[6], Daniel Aharoni[4], Scott A Wilke[5], Laura A DeNardo[1]*

[1]Department of Physiology, University of California, Los Angeles, Los Angeles, United States; [2]UCLA Neuroscience Interdepartmental Program, University of California, Los Angeles, Los Angeles, United States; [3]UCLA Molecular, Cellular, and Integrative Physiology Program, University of California, Los Angeles, Los Angeles, United States; [4]Department of Neurology, University of California, Los Angeles, Los Angeles, United States; [5]Department of Psychiatry, University of California, Los Angeles, Los Angeles, United States; [6]Department of Psychology, University of California, Los Angeles, Los Angeles, United States

*For correspondence:
ldenardo@ucla.edu

**Abstract** Quantitative descriptions of animal behavior are essential to study the neural substrates of cognitive and emotional processes. Analyses of naturalistic behaviors are often performed by hand or with expensive, inflexible commercial software. Recently, machine learning methods for markerless pose estimation enabled automated tracking of freely moving animals, including in labs with limited coding expertise. However, classifying specific behaviors based on pose data requires additional computational analyses and remains a significant challenge for many groups. We developed BehaviorDEPOT (DEcoding behavior based on POsitional Tracking), a simple, flexible software program that can detect behavior from video timeseries and can analyze the results of experimental assays. BehaviorDEPOT calculates kinematic and postural statistics from keypoint tracking data and creates heuristics that reliably detect behaviors. It requires no programming experience and is applicable to a wide range of behaviors and experimental designs. We provide several hard-coded heuristics. Our freezing detection heuristic achieves above 90% accuracy in videos of mice and rats, including those wearing tethered head-mounts. BehaviorDEPOT also helps researchers develop their own heuristics and incorporate them into the software's graphical interface. Behavioral data is stored framewise for easy alignment with neural data. We demonstrate the immediate utility and flexibility of BehaviorDEPOT using popular assays including fear conditioning, decision-making in a T-maze, open field, elevated plus maze, and novel object exploration.

## Editor's evaluation

In this manuscript, the authors introduce a new piece of software, BehaviorDEPOT, that aims to serve as an open-source classifier in service of standard lab-based behavioral assays. The key arguments the authors make are that (1) the open-source code allows for freely available access, (2) the code doesn't require any coding knowledge to build new classifiers, (3) it is generalizable to other behaviors than freezing and other species (although this latter point is not shown), (4) that it uses posture-based tracking that allows for higher resolution than centroid-based methods, and (5) that it

is possible to isolate features used in the classifiers. These aims are laudable, the software is indeed relatively easy to use, and it will likely be used widely in the community.

## Introduction

A central goal of neuroscience is to discover relationships between neural activity and behavior. Discrete behaviors represent outward manifestations of cognitive and emotional processes. Popular laboratory assays for memory, decision-making, anxiety and novelty exploration typically require examination of behaviors occurring in particular locations in space or instances in time. However, it remains a major challenge to quantify naturalistic behaviors in the laboratory in a rapid, reliable, and spatiotemporally precise manner. In most labs, this work is done manually, a time-consuming process that is rife with inconsistency.

Automated detection of freely moving animal behaviors is faster and expands the parameter space that can be explored. Automated detection can also eliminate errors associated with manual annotation such as inter-rater inconsistency due to insufficient rater training and rater fatigue. The standardization promised by such methods enhances the rigor and reproducibility of results across research groups, which is a major concern in behavioral neuroscience. Commercially available software for automated behavior analysis is expensive and the underlying algorithms are hidden, which prevents customization or interrogation to determine why a particular result is reported. Moreover, commercially available behavior detectors are prone to failure when animals are wearing head-mounted hardware for manipulating or recording brain activity. As open-source hardware for recording and manipulating neural activity become increasingly available (*Luo et al., 2018*), more labs are integrating optogenetics, miniscopes, fiber photometry and electrophysiological recordings into their behavioral experiments. This expansion of the research space includes labs without established computational expertise to quantify complex behaviors or align them with precisely timed manipulations or biological signals. Flexible, easy-to-use, open-source software is needed to automate analysis of freely moving behaviors and facilitate subsequent analyses.

To label behaviors automatically, animals must first be segmented from their environment and tracked through space and time. Previously established methods use techniques including background subtraction and pattern classifiers to estimate animal positions in video timeseries and then abstract animal movement to a center of mass or an ellipse for analysis (*Ohayon et al., 2013*; *Branson et al., 2009*; *Noldus et al., 2001*; *Geuther et al., 2021*). After segmentation and tracking, the challenge then is to use data about animal movements to classify discrete behaviors that represent useful information about the cognitive or emotional state of the animal. JAABA (*Kabra et al., 2013*) uses machine learning to classify behaviors based on the outputs of tracking systems such as MotionTracker (MoTr) (*Ohayon et al., 2013*) and Ctrax (*Branson et al., 2009*), which fit ellipses to track animal movements. JAABA requires no coding expertise and has been widely used in the fly community to report complex behaviors including social interactions and various kinds of locomotion. However, only a few rodent studies have employed JAABA (*Sangiamo et al., 2020*; *Neunuebel et al., 2015*; *Phillips et al., 2019*; *Nomoto and Lima, 2015*; *van den Boom et al., 2017*). One limitation is that ellipses cannot resolve the detailed spatiotemporal relationships between individual body parts that characterize many complex behaviors. Moreover, many methods that rely on background subtraction or similar approaches are not robust to environmental complexity and require behavioral assays that can be performed in an empty arena (*Geuther et al., 2019*).

Newer pose estimation algorithms that are based on machine learning can accurately track individual 'keypoints' on an animal's body (e.g. nose, ears, joints) (*Mathis et al., 2018*; *Pereira et al., 2019*; *Graving et al., 2019*). Keypoint tracking is robust to environmental changes in behavioral arenas and the relationships between the locations of keypoints allow researchers to resolve temporal pose sequences with fine detail. Recently developed open-source software packages such as MARS (*Segalin et al., 2021*) and SimBA (*Nilsson et al., 2020*) use supervised machine learning to classify behaviors based on the positions of keypoints on an animal's body. These methods are excellent solutions for classifying complex behaviors that are challenging for humans to label reliably, including multi-animal social behaviors or self-grooming. However, additional rounds of machine learning are computationally-intensive and require significant amounts of human-labeled video data. This level of complexity is unnecessary to classify many widely studied behaviors and implementing these

approaches may present an insurmountable challenge for labs with limited expertise in coding and machine learning.

Here we describe BehaviorDEPOT, which provides an easy way to convert keypoint tracking into meaningful behavioral data in a wide variety of experimental configurations. BehaviorDEPOT not only detects behaviors in video timeseries but can analyze the results of widely used assays such as fear conditioning, open field test, elevated plus maze, novel object exploration, and T-mazes, and can accommodate varied designs including optogenetic manipulations. In these assays, behaviors of interest are typically defined based on criteria set by individual researchers. Definitions often include both the pose of the animal and a physical location in the behavioral arena. For example, in novel object exploration, bouts of exploration are typically defined as times when the animal's head is oriented towards the object and the animal is within 2 cm of the object. Keypoint tracking is ideally suited for these types of behaviors because it can track the spatial location of animals while also resolving fine-scale movements.

BehaviorDEPOT detects behaviors using heuristics – simple, efficient rules – that are based on human definitions. These heuristics operate by applying thresholds to metrics calculated from keypoint tracking data (e.g. velocity, angular velocity), and can also incorporate spatial and temporal cues of the experimenters' choosing. BehaviorDEPOT heuristics have low error rates and, in contrast to classifiers built through additional rounds of machine learning, can be developed based on small numbers of manually annotated video frames and can be easily tweaked to fit out-of-sample videos. Our freezing heuristic has excellent performance even in animals wearing tethered patch cords for optogenetics or $Ca^{2+}$ imaging, thereby overcoming a major point of failure in commercially available freezing algorithms. BehaviorDEPOT organizes and saves behavioral data in structures that facilitate subsequent analyses, including alignment with neural recordings. It also helps users develop their own heuristics and incorporate them into the graphical interface. The automated, intuitive, and flexible way in which BehaviorDEPOT quantifies behavior will propel new discoveries by allowing even inexperienced coders to capitalize on the richness of their data.

## Results

BehaviorDEPOT comprises six independent modules that form a flexible, multifunctional pipeline that can run experiments, detect behaviors in video timeseries, and analyze behavioral data (*Figure 1*). Its graphical interface accommodates users with no prior coding experience. The Analysis Module imports keypoint tracking data, calculates postural and kinematic metrics (e.g. body length, head velocity), uses these data as the basis of heuristic behavior detectors, and analyzes the results of experiments (e.g. report the effects of an optogenetic manipulation during cued fear conditioning). We provide hard-coded heuristics for detecting freezing, jumping, rearing, escape, locomotion, and novel object exploration. We also provide analysis functions for the open field test, elevated plus maze, T-maze, and three chamber assays. The Analysis Module generates rich data structures containing spatial tracking data, postural and kinematic data, behavioral timeseries, and experimental parameters. All data is stored framewise for easy alignment with neural signals. We developed the Experiment Module as a companion to our heuristic for detecting freezing. The Experiment Module can run fear conditioning experiments by using Arduinos to control shockers, arenas, and lasers for optogenetics. Finally, to maximize the utility and flexibility of BehaviorDEPOT, we created the Inter-Rater, Data Exploration, Optimization and Validation Modules to guide users through developing their own heuristics and integrating them into the Analysis Module (*Figure 1*). Here, we describe how each module works and demonstrate BehaviorDEPOT's utility in fear conditioning, learned avoidance, open field, elevated plus maze, novel object exploration assays, and in an effort-based decision task in a T-maze.

### The analysis module

The main functions of the Analysis Module are to automatically detect behaviors and analyze the results of experiments. The Analysis Module imports videos and accompanying keypoint tracking data and smooths the tracking data. It can accommodate previously recorded videos since keypoint tracking models can be trained posthoc. Users can track any keypoints they choose. We used DeepLabCut (DLC) (*Mathis et al., 2018*) for keypoint tracking. DLC produces a list of comma-separated

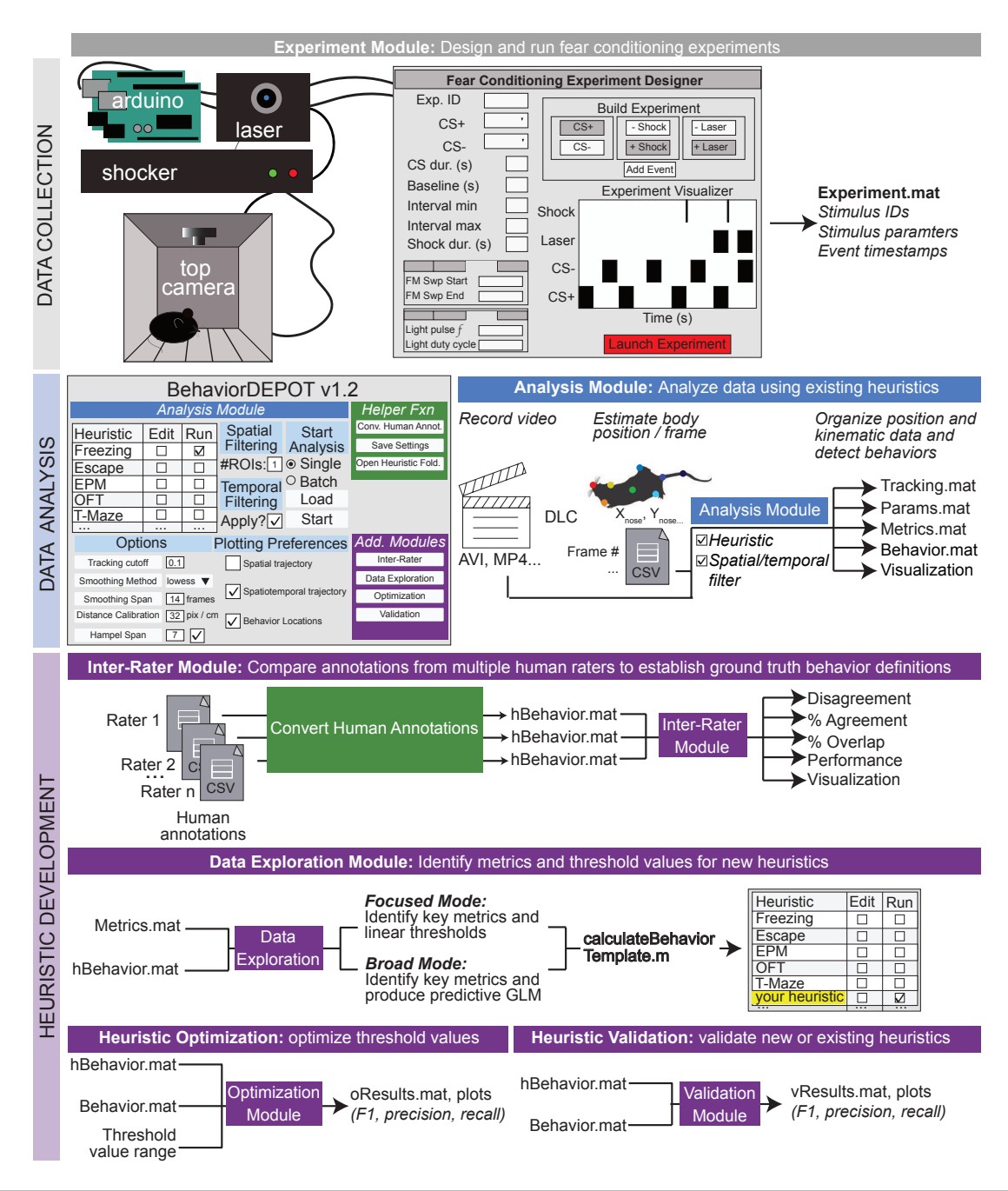

**Figure 1.** BehaviorDEPOT is a general-purpose behavioral analysis software comprising six modules. The Experiment Module is a MATLAB application with a graphical interface that allows users to design and run fear conditioning experiments. The software uses Arduinos to interface with commercially available hardware (e.g. shockers and lasers) to control stimuli. The Analysis Module imports keypoint tracking data, calculates postural and kinematic metrics, detects behaviors, and analyzes the results of behavior experiments. Four additional modules help users develop custom heuristics. The Inter-Rater Module compares human annotations, helping researchers settle on 'ground truth' definitions of behavior. These human definitions serve as reference data for behavioral detection heuristics. The data exploration module identifies features of movement with the highest predictive value for specific behaviors. The optimization module identifies combinations of feature thresholds that maximize behavioral detection. The validation module reports the accuracy of heuristics.

The online version of this article includes the following figure supplement(s) for figure 1:

**Figure supplement 1.** Example arrangement of Arduino interface between computer, fear conditioning, and optogenetics hardware.

values that contains framewise estimates of the X-Y coordinates for designated body parts as well as a likelihood statistic for each estimated point. The Analysis Module applies a threshold based on DLC likelihood and performs a Hampel transformation (*Hampel, 1974*) to remove outliers. Then, a local regression smoothing method (LOWESS) is applied to the data (*Cleveland, 1981*), and sub-threshold tracking values are estimated using surrounding data and spline interpolation (*Figure 2A*). The module then performs a feature expansion step, calculating additional keypoint positions based on the originally tracked set (*Supplementary file 1*). We designed a set of postural and kinematic metrics that are calculated automatically for each keypoint (*Supplementary file 2*). These metrics serve as the inputs for BehaviorDEPOT's heuristics.

We created a set of heuristics to detect several human-defined behaviors including freezing, rearing, escape, locomotion, and novel object investigation. From the graphical interface of the Analysis Module, users can select a heuristic and indicate if they want to analyze behaviors during particular time windows or within regions of interest (ROI) and whether they plan to do batched analyses. The Analysis Module also includes spatial functions for analyzing the open field test, elevated plus maze, T-maze and three chamber assays. By allowing users to perform integrated spatial-temporal-behavioral analyses, BehaviorDEPOT's utility extends beyond existing open-source behavioral classification software, which typically just report which behaviors occur in each video frame. Later in the manuscript, we describe how users can create their own heuristics and incorporate them into the graphical interface.

The Analysis Module exports all relevant data in a set of MATLAB structures (*Supplementary file 3*) so that users can easily perform additional analyses if needs arise (*Figure 2A*). For instance, users may want to align the data to neural signals, a process we discuss in upcoming sections. A structure called 'Tracking' stores raw and smoothed keypoint tracking data. 'Params' stores the parameters of video recordings, smoothing functions, heuristics used, and arena metrics (e.g. ROI size and location). 'Metrics' stores kinematic and postural statistics calculated from keypoint positions (*Supplementary file 2*). 'Behavior' stores bout-wise and vectorized representations of identified behaviors across the entire video, or with respect to user-defined spatiotemporal filters.

The Analysis Module also generates a series of graphical data representations (*Supplementary file 3*). For instance, trajectory maps show when an animal was in a particular location and where behaviors occurred. Bout maps indicate when behaviors occurred and for how long. These visual representations help users understand behavioral phenotypes in great spatiotemporal detail and can inform further custom analyses using the data structures that the Analysis Module generates. In the following sections, we describe the development and validation of BehaviorDEPOT's heuristics and demonstrate its utility in numerous behavioral assays.

## Development and validation of the BehaviorDEPOT freezing detection heuristics

Freezing behavior, defined as the absence of movement except for respiration (*Grossen and Kelley, 1972*; *Fanselow and Bolles, 1979*; *Fanselow, 1984*), is widely studied as a proxy for both learned and innate fear (*Anagnostaras et al., 1999*; *Anagnostaras et al., 2010*; *Perusini and Fanselow, 2015*). In many laboratory assays, rodents freeze in response to perceived environmental threats including conditioned cues, predator odor, or looming disks that mimic a predator approaching from above (*Perusini and Fanselow, 2015*). Despite the heavy study of freezing behavior, most labs still score freezing manually or using expensive commercially available software programs that often fail when animals have tethered headmounts. Here, we describe the development and validation of the BehaviorDEPOT freezing detection heuristic and demonstrate its accurate performance across a range of experimental setups, including in mice wearing head-mounts for optogenetics and Miniscopes.

The BehaviorDEPOT freezing detection heuristic combines a convolution-based smoothing operation with a low-pass velocity filter and then applies a minimum duration threshold to identify periods of freezing. To develop the freezing heuristic, we began by training a deep neural network in DLC on videos recorded with a high resolution and high frame rate camera (Chameleon3 USB3, FLIR) at 50 frames per second (fps). The network tracked 8 points on the body and BehaviorDEPOT smoothed the raw tracking data (*Figure 2B and C*). An expert rater annotated freezing in three randomly selected videos (27,000 total frames). This served as a reference set for heuristic development. We reasoned that freezing could be reliably detected based on periods of low keypoint velocity. After exploring

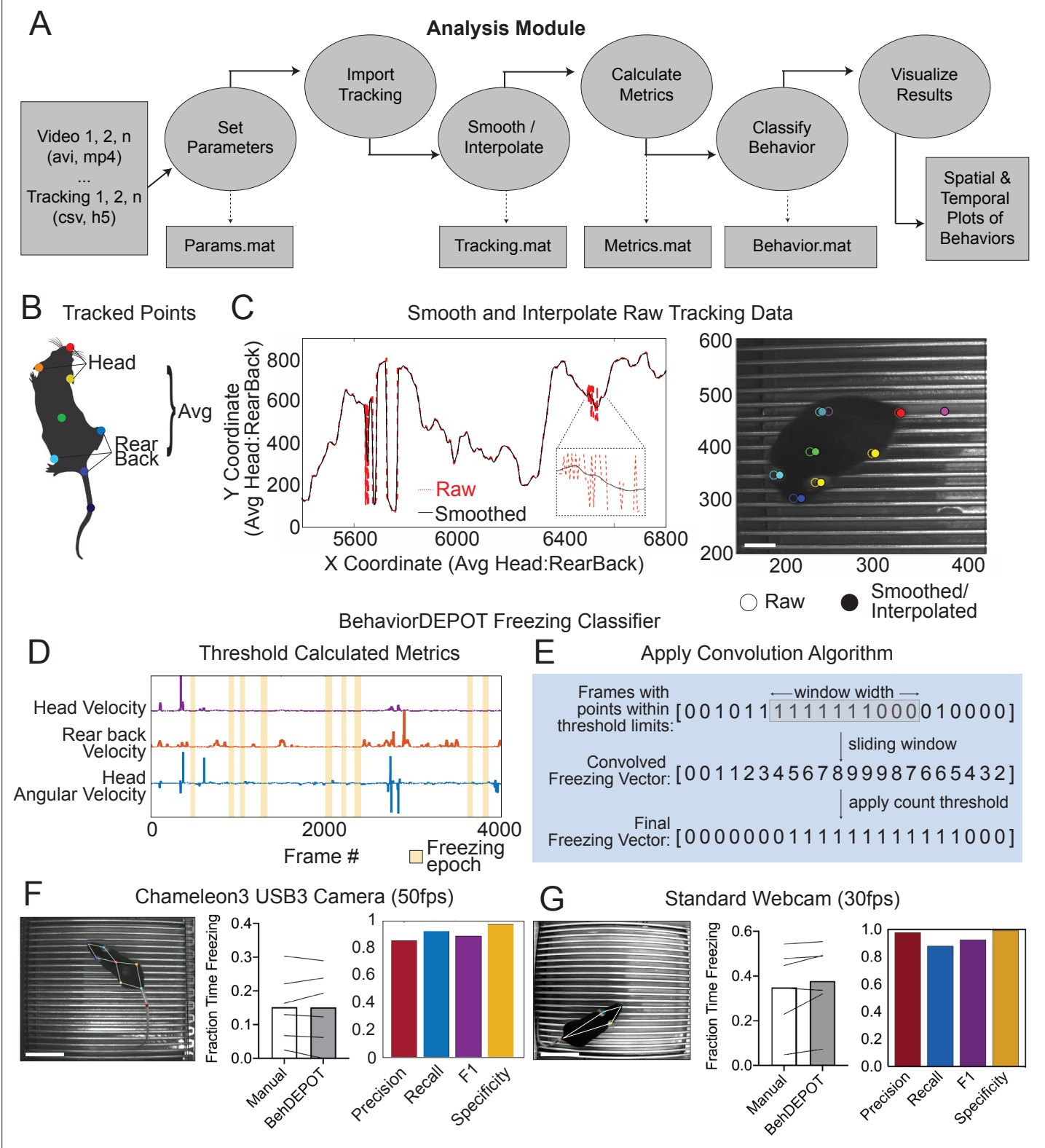

**Figure 2.** The Analysis Module. (**A**) The Analysis Module workflow. Videos and accompanying pose tracking data are the inputs. Pose tracking and behavioral data is vectorized and saved in MATLAB structures to facilitate subsequent analyses. (**B**) Metrics based on individual tracked points and weighted averages are calculated and stored in BehaviorDEPOT data matrices. (**C**) Visualization of the effects of the LOWESS smoothing and interpolation algorithms for the weighted average of head and rear back (left) and for all tracked points in a representative example frame (right; scale

*Figure 2 continued on next page*

*Figure 2 continued*

bar, 2 cm). (**D**) Visualization of metrics that form the basis of the BehaviorDEPOT freezing heuristic. Colored lines represent framewise metric values. Yellow bars indicate freezing epochs. (**E**) Visualization of the convolution algorithm employed by the BehaviorDEPOT freezing heuristic. A sliding window of a specified width produces a convolved freezing vector in which each value represents the number of freezing frames visible in the window at a given frame. An adjustable count threshold converts the convolved freezing vector into the final binary freezing vector. (**F**) Evaluation of freezing heuristic performance on videos recorded at 50 fps with a high resolution, high framerate camera ($p$=0.95, paired t-test, N=6; *Precision*: 0.86, Recall: 0.92, F1: 0.89, Specificity: 0.97). Scale bar, 5 cm. (**G**) Evaluation of freezing heuristic performance on videos recorded at 30fps with a standard webcam ($p$=0.10, paired t-test, N=6; Precision: 0.98, Recall: 0.88, F1: 0.93, Specificity: 0.99). Scale bar, 5 cm.

The online version of this article includes the following figure supplement(s) for figure 2:

**Figure supplement 1.** Performance of the freezing heuristic based on DLC mean tracking error.

**Figure supplement 2.** Performance of the 'Jitter' Freezing Heuristic on Webcam videos.

the predictive value of different keypoints using the Data Exploration Module (described below), we determined that thresholding the velocity of a midpoint on the back and the angular velocity of the head produced the most accurate freezing detection heuristic (*Figure 2D*). We smoothed the heuristic output by applying a sliding window to produce a convolved freezing vector in which each value represented the number of freezing frames visible when the window is centered at a given frame. We then applied an adjustable count threshold to convert the convolved freezing vector into the final binary freezing vector (*Figure 2E*).

To validate heuristic performance, we manually annotated a different, randomly selected set of videos that were never referenced while setting the parameters (*Supplementary file 4*). These videos were recorded in several different behavioral chambers under varied lighting conditions. Behavior-DEPOT freezing detection was highly consistent with human annotations (*Figure 2F*). Accuracy of the freezing heuristic was estimated based on precision, recall, F1 score, and specificity. Precision and recall quantify the positive predictive value against the tendency to produce false positive or false negative errors, respectively. The F1 score, the harmonic mean of the precision and recall, is useful as a summary statistic of overall performance. Specificity quantifies the ability to accurately label true negative values and helps ensure that the heuristic is capturing data from only a single annotated behavior. Our heuristic had very low error rates (*Figure 2F*).

We also assessed how good the DLC tracking needed to be for the freezing heuristic to work well. We trained ten different DLC networks with mean tracking errors ranging from 1 to 8 pixels. We used each network to analyze a set of six videos and then used BehaviorDEPOT to automatically detect freezing in each video. A linear regression analysis revealed that tracking error had a significant effect on precision and on the F1 score without affecting recall or specificity (*Figure 2—figure supplement 1*; *Supplementary file 5*), with tracking errors <4 pixels producing the highest F1 scores in our analysis.

To ensure that our heuristic would generalize to other camera types and DLC networks, we trained a second network based on videos recorded with a standard webcam at 30fps. On the webcam, lower sensor quality and lower frame rate produces more blur in the recorded images, so we tracked 4 body parts that are easy to see (nose, ears, tail base; *Figure 2G*). When compared to human annotations (*Supplementary file 4*), the webcam videos also scored highly for precision, recall, F1 score, and specificity (*Figure 2G*), indicating that our freezing heuristic can indeed generalize across camera types and DLC networks.

Since the same rules may not necessarily generalize to all settings, we developed a second freezing heuristic that uses a changepoint function to find frames at which the mean velocity changes most significantly and then separates frames into groups that minimize the sum of the residual error from the local mean. Binarized freezing vectors are then processed using a convolution algorithm and minimum duration threshold (*Figure 2E*). This heuristic was also highly accurate, performing similarly to the velocity-based heuristic on videos recorded with a webcam (*Figure 2—figure supplement 2*). Users can tune the heuristic by adjusting a minimum residual threshold for the changepoint function. We termed this the 'jitter' heuristic since the minimum residual threshold will be determined by the pixel error in keypoint position estimates. In other words, the threshold will be determined by how much frame-to-frame 'jitter' there is in DLC's estimate of the keypoint location. Keypoint tracking 'jitter' may arise as a function of video resolution, framerate, and number of frames used to train a keypoint tracking model. As such, the 'jitter' heuristic may accommodate a wider range of video

qualities and keypoint tracking models. Also, in videos recorded from the side, the velocity-based freezing heuristic may be slightly affected by distortions in velocity calculations caused by the angle (e.g. when the mouse is moving towards/away from the camera).

## The experiment module

As a companion to the heuristic for freezing, we also developed the 'Experiment Module', a MATLAB app that allows users to design and run fear conditioning experiments. This extends the utility of BehaviorDEPOT by providing a fully open-source software pipeline that takes users from data collection to data analysis. The Experiment Module controls commercially available shockers, lasers for optogenetics, and sound generators via a set of Arduinos (*Figure 1—figure supplement 1*). Users can download the 'Fear Conditioning Experiment Designer' app from our Github repository and install it with a single button click. From a graphical interface, users can design experimental protocols for contextual or cued fear conditioning, with or without optogenetics. All experimental parameters (e.g. timestamps for laser, tones, shocks) are saved in MATLAB structures that can be referenced by the BehaviorDEPOT analysis pipeline (*Figure 1*).

## Use Case 1: Optogenetics

In commercial freezing detection software, algorithms often fail when a rodent is wearing a patch cord for routine optogenetics experiments. To ensure that BehaviorDEPOT's freezing heuristic maintains high levels of accuracy under these conditions, we tested its performance in an optogenetics experiment. mPFC plays a well-established role in fear memory retrieval (*Corcoran and Quirk, 2007*; *LeDoux, 2000*), extinction (*Giustino and Maren, 2015*; *Sierra-Mercado et al., 2011*), and generalization (*Giustino and Maren, 2015*; *Xu and Südhof, 2013*; *Pollack et al., 2018*). While silencing mPFC subregions can promote fear memory generalization in remote memory (*Xu and Südhof, 2013*; *Frankland et al., 2004*), less is known about its role in recent memory generalization. We used an adeno-associated virus (AAV) to express the soma-targeted neuronal silencer stGtACR2 (*Mahn et al., 2018*) bilaterally in the mPFC and implanted optogenetic cannula directly above the AAV injection sites (*Figure 3A*, *Figure 3—figure supplement 1*). We performed contextual fear conditioning (CFC) in context A. The next day, we measured freezing levels in the conditioned context (context A) as well as a novel context (context B) that was never paired with shocks. During these retrieval sessions, 2-min laser-on periods were separated by two minute laser-off intervals (*Figure 3B*).

We first tested the accuracy of the freezing heuristic for animals wearing tethered head mounts. We trained an optogenetics-specific DLC network that tracks 9 points on the animal, including the fiber-optic cannula (*Figure 3C*). Our rationale for creating a separate keypoint tracking network was twofold. First, while you can train one 'master' DLC network that can track mice in many different arenas, we find that DLC tracking errors are lowest when you have dedicated networks for particular camera heights, arenas, and types of head-mounted hardware. DLC networks are easy to train and to make it even easier for new users, we provide links to download our DLC models in our GitHub repository. Second, this was another opportunity to test how well the BehaviorDEPOT freezing heuristic generalizes to a different DLC network with a different number of keypoints. For a randomly selected set of videos with different floors and lighting conditions, we compared heuristic performance to expert human raters (*Supplementary file 4*). Even with the patch cord attached, the freezing heuristic had excellent scores for precision, recall, F1 and specificity (*Figure 3D*).

Having confirmed the accuracy of the freezing heuristic, we then used BehaviorDEPOT to quantify freezing behavior and compared the results to the annotations of an expert human rater. As expected, fear conditioned mice readily froze following shocks during CFC, while non-shocked controls did not (*Figure 3E*). During retrieval sessions, silencing mPFC in previously shocked animals significantly enhanced freezing in the novel context but did not affect freezing in the fear conditioned context (*Figure 3F and G*). mPFC silencing thereby produced a significant decrease in the discrimination index in fear conditioned mice (*Figure 3H*), indicating that mPFC plays a key role in the specificity of recent fear memories. In all analyses, BehaviorDEPOT freezing estimates were comparable to a highly trained human rater (*Figure 3F–H*). By maintaining performance levels even in videos with visual distractors like a patch cord, the BehaviorDEPOT freezing heuristic overcomes a major point of failure in commercially available freezing detection software.

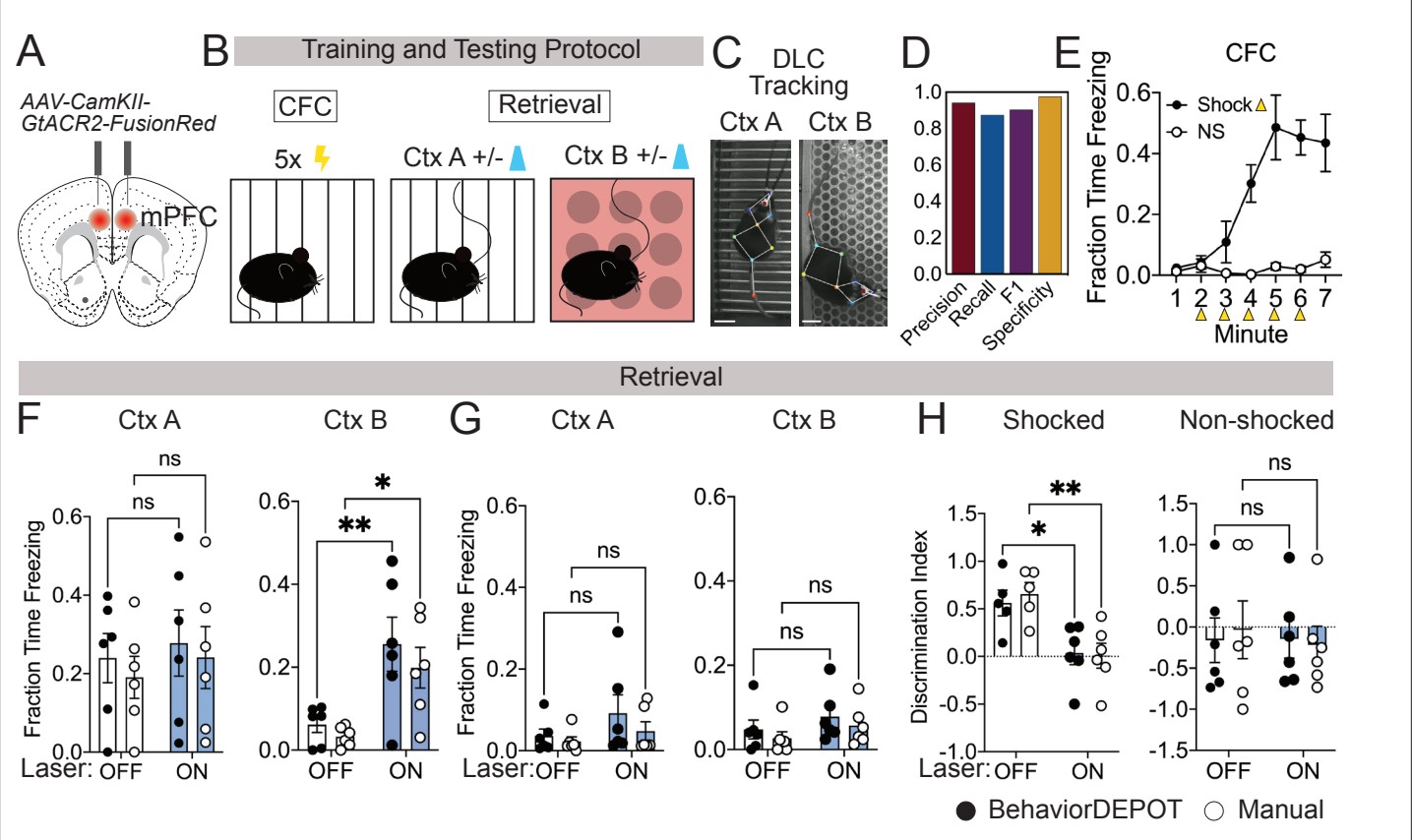

**Figure 3.** Use Case 1: Optogenetics. (**A**) AAV1-CamKII-GtACR2-FusionRed was injected bilaterally into medial prefrontal cortex (mPFC). (**B**) Behavioral protocol. Mice underwent contextual fear conditioning on day 1. On day 2, mice were returned to the conditioned context or a novel context in a counterbalanced fashion and received 2x2 min 473 nm laser stimulation separated by 2 min laser off intervals. (**C**) Example DLC tracking of mice attached to patch cords in different contexts. Scale bars, 2cm. (**D**) Performance of freezing heuristic (Precision: 0.94, Recall: 0.87, F1: 0.91, Specificity: 0.98). (**E**) Quantification of contextual freezing during training analyzed with BehaviorDEPOT. F–G. Comparing human annotations to BehaviorDEPOT freezing heuristic. (**F**) Shocked mice: freezing in context A (left) and context B (right) with and without mPFC silencing (CtxA: $F_{laser}(1,10)=0.42$, p=0.53; $F_{rater}(1,10)=0.35$, p=0.57; Off vs. On: $P_{BD} = 0.91$, $P_{Manual} = 0.86$; CtxB: $F_{laser}(1,10)=26.51$, p=0.0004; $F_{rater}(1,10)=0.08$, p=0.78; Off vs. On: $P_{BD} = 0.008$, $P_{Man} = 0.02$; Two-way repeated measures ANOVA and Sidak's test, N=6 mice per group). (**G**) Non-shocked controls: freezing in context A (left) and context B (right) with and without mPFC silencing (Ctx A: $F_{laser}(1,10)=3.60$, p=0.09; $F_{rater}(1,10)=0.79$, p=0.39; Off vs. On: $P_{BD} = 0.30$, $P_{Manual} = 0.76$; CtxB: $F_{laser}(1,10)=1.486$, $P=0.25$; $F_{rater}(1,10)=1.59$, p=0.24; Off vs. On: $P_{BD} = 0.52$, $P_{Manual} = 0.54$; Two-way repeated measures ANOVA, N=6 mice per group). (**H**) Discrimination index = (FreezeA - FreezeB) / (FreezeA +FreezeB) for shocked mice ($F_{laser}(1,10)=17.54$, p=0.002; $F_{rater}(1,8)=0.09$, p=0.77; Mixed-effects analysis, On vs. Off: $P_{BD} = 0.02$, $P_{Manual} = 0.004$, Sidak's test, N=5–6 per group) and non-shocked controls ($F_{laser}(1,10)=0.07$, p=0.80; $F_{rater}(1,8)=0.02$, p=0.90; Two-way ANOVA; On vs. Off: $P_{BD} = ,0.99$ $P_{Manual} = 0.87$, Sidak's test, N=6 per group). Error bars represent S.E.M.

The online version of this article includes the following figure supplement(s) for figure 3:

**Figure supplement 1.** Histology for optogenetics viral injections and fiber implants.

## Use Case 2: Ca²⁺ imaging with Miniscopes during signaled avoidance

As new open-source tools for neurophysiology become available, more labs are performing simultaneous neurophysiological and behavioral recordings. Miniature head-mounted microscopes now allow us to image the activity of hundreds of neurons simultaneously in freely moving animals (*Ghosh et al., 2011*; *Cai et al., 2016*; *Shuman et al., 2020*). These miniscopes pair with genetically encoded Ca²⁺ indicators (*Dana et al., 2019*) that can be targeted to specific neuronal populations and GRIN lenses (*Barretto et al., 2009*) that can be targeted anywhere in the brain. With these tools in hand, we can discover how the encoding of complex cognitive and emotional behaviors maps onto specific cell types across the brain. By recording the activity of hundreds of neurons simultaneously, we can also study the population codes that produce complex behaviors (*Jercog et al., 2021*; *Stout and Griffin, 2020*). To do so, however, we need improved open-source methods that allow us to quantify

freely moving behaviors with reference to salient environmental stimuli and to align these detailed behavioral measurements with neurophysiological recordings.

Here, we demonstrate the utility of BehaviorDEPOT for aligning behavioral measurements with $Ca^{2+}$ signals during a platform-mediated avoidance (PMA) task that has temporally and spatially salient features. PMA is an mPFC-dependent task in which a fear conditioned tone prompts mice to navigate to a safety platform that protects them from receiving a footshock (*Bravo-Rivera et al., 2015*; *Diehl et al., 2020*). We recorded the activity of hundreds of mPFC neurons in freely behaving animals using head-mounted microendoscopes (UCLA Miniscopes *Cai et al., 2016*; *Shuman et al., 2020*) while simultaneously recording behavior using a new open-source USB camera, the UCLA MiniCAM.

Together with BehaviorDEPOT and the UCLA Miniscopes, the MiniCAM provides a fully open-source data acquisition and analysis pipeline for in vivo $Ca^{2+}$ imaging during freely moving behavior. The MiniCAM is an open-source behavioral imaging platform that natively integrates and synchronizes with open-source UCLA Miniscope hardware and software (*Figure 4A*). It is composed of an M12 optical lens mount, a custom printed circuit board housing a CMOS image senor and supporting electronics, an LED illumination ring, and a 3D printed case. The MiniCAM is powered and communicates over a single coaxial cable that can be up to 15 m long. The coaxial cable connects to a Miniscope data acquisition board (DAQ) which then connects over USB3 to a host computer. A range of commercial M12 lenses can be used to select the view angle of the camera system. The image sensor used is a 5MP CMOS image sensor (MT9P031I12STM-DP, ON Semiconductor) with 2592x1944 pixel resolution and a full resolution frame rate of approximately 14FPS. For this application, the MiniCAM's pixels were binned and cropped to achieve 1024x768 pixels at approximately 50FPS. The optional LED illumination ring uses 16 adjustable red LEDs (LTST-C190KRKT, Lite-On Inc, 639 nm peak wavelength) for illumination in dark environments (*Figure 4A*). We trained a separate DLC network for videos of animals wearing Miniscopes recorded with MiniCAMs. Our network tracked 8 keypoints on the body (ears, nose, midback, hips, tailbase, and tail) and the Miniscope itself (*Figure 4B*). For these videos, BehaviorDEPOT freezing ratings were highly consistent with expert human annotations (*Figure 4C*).

We used BehaviorDEPOT to analyze behavior during PMA so we could align it to the underlying $Ca^{2+}$ activity. In this task, animals are placed in a fear conditioning chamber in which an acrylic safety platform occupies 25% of the electrified grid floor. Three baseline tones are followed by nine tones that co-terminate with a mild foot shock. The following day, we measure avoidance and freezing behaviors during six unreinforced tones (*Figure 4D*). The Analysis Module automatically produces trajectory maps that make it quick and easy to assess the spatiotemporal characteristics of rodent behavior. In our representative example, the color-coded trajectory and freezing locations (denoted as black squares) converge on the platform at the end of the session, indicating the mouse indeed learned to avoid shocks by entering the platform (*Figure 4E*). We used BehaviorDEPOT to produce summary data, showing that mice readily learned the cue-avoidance association during training (*Figure 4F*) and remembered it the next day (*Figure 4G*).

During a retrieval session, we recorded $Ca^{2+}$ activity using a UCLA Miniscope and behavior using a UCLA MiniCAM (*Figure 4H*). Using MIN1PIPE (*Lu et al., 2018*), we extracted and processed neural signals from 513 mPFC neurons across 3 mice. We then determined whether individual neurons encoded specific behaviors that we had quantified using BehaviorDEPOT (*Figure 4I*). We computed a receiver operating characteristic (ROC) curve that measures a neuron's stimulus detection strength over a range of thresholds (*Figure 4J*). We identified numerous neurons that were modulated by freezing and avoidance on the safety platform. These neurons were organized in a salt and pepper manner in mPFC (*Figure 4K*). Nearly half of all neurons that exhibited task relevant behavior and were specific for either freezing or threat avoidance, or their combination (*Figure 4L*). These experiments demonstrate that the BehaviorDEPOT heuristic for detecting freezing is robust across a wide variety of experimental setups with different camera types, keypoint tracking networks, arenas and head-mounts. Further, this suite of open-source hardware and software will enable a broader user base to combine $Ca^{2+}$ imaging with high resolution behavioral analysis.

## Use Cases 3–6: open field, elevated plus maze, novel object exploration, and decision-making

BehaviorDEPOT also supports behavioral analyses beyond the realm of conditioned fear. In a number of commonly used assays, behaviors are defined based on animal location or the intersection of

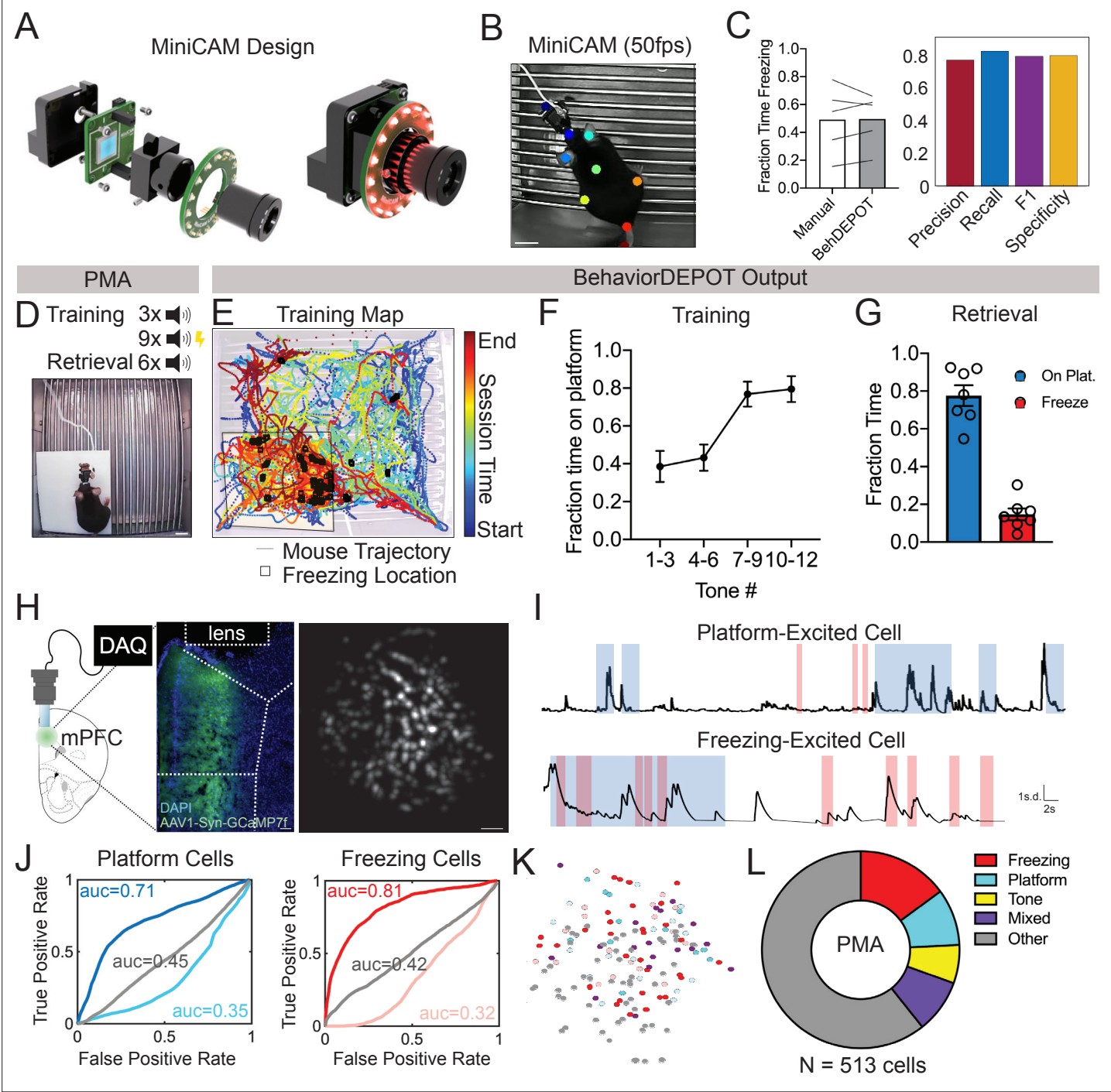

**Figure 4.** Use Case 2: Mice wearing Miniscopes. (**A**) Design for MiniCAM, an open-source camera designed to interface with Miniscopes and pose tracking. (**B**) Still frame from MiniCAM recording of mouse wearing a V4 Minscope. DLC tracked points are labeled with rainbow dots. Scale bar, 2 cm. (**C**) Performance of freezing heuristic on videos of mouse wearing Miniscope recorded with MiniCAM (Precision: 0.85; Recall: 0.93; F1 Score: 0.89; Specificity: 0.84). (**D**) Task design. (**E**) Sample BehaviorDEPOT output for mouse wearing Miniscope during PMA. Map displays animal position over time as well as freezing locations (black squares). (**F**) Summary behavioral data for training. (**G**) Summary behavioral data for retrieval. (**H**) GCaMP7-exressing mPFC neurons imaged through a V4 Miniscope. Scale bars, 100 μm. (**I**) Example $Ca^{2+}$ traces from platform (top) and tone (bottom) modulated cells during time on the platform (blue) or time freezing (pink). (**J**) Receiver operating characteristic (ROC) curves that were calculated for platform-modulated cells (excited cell: auc = 0.71; suppressed cell: auc = 0.35, unmodulated cell: auc = 0.45) and freezing-modulated cells (excited cell: auc = 0.81; suppressed cell: auc = 0.32; unmodulated cell: auc = 0.42). (**K**) Example field of view showing locations of freezing- and platform-modulated mPFC neurons. (**L**) Proportion of modulated cells of each functional type from 513 cells recorded across 3 mice. Error bars represent S.E.M.

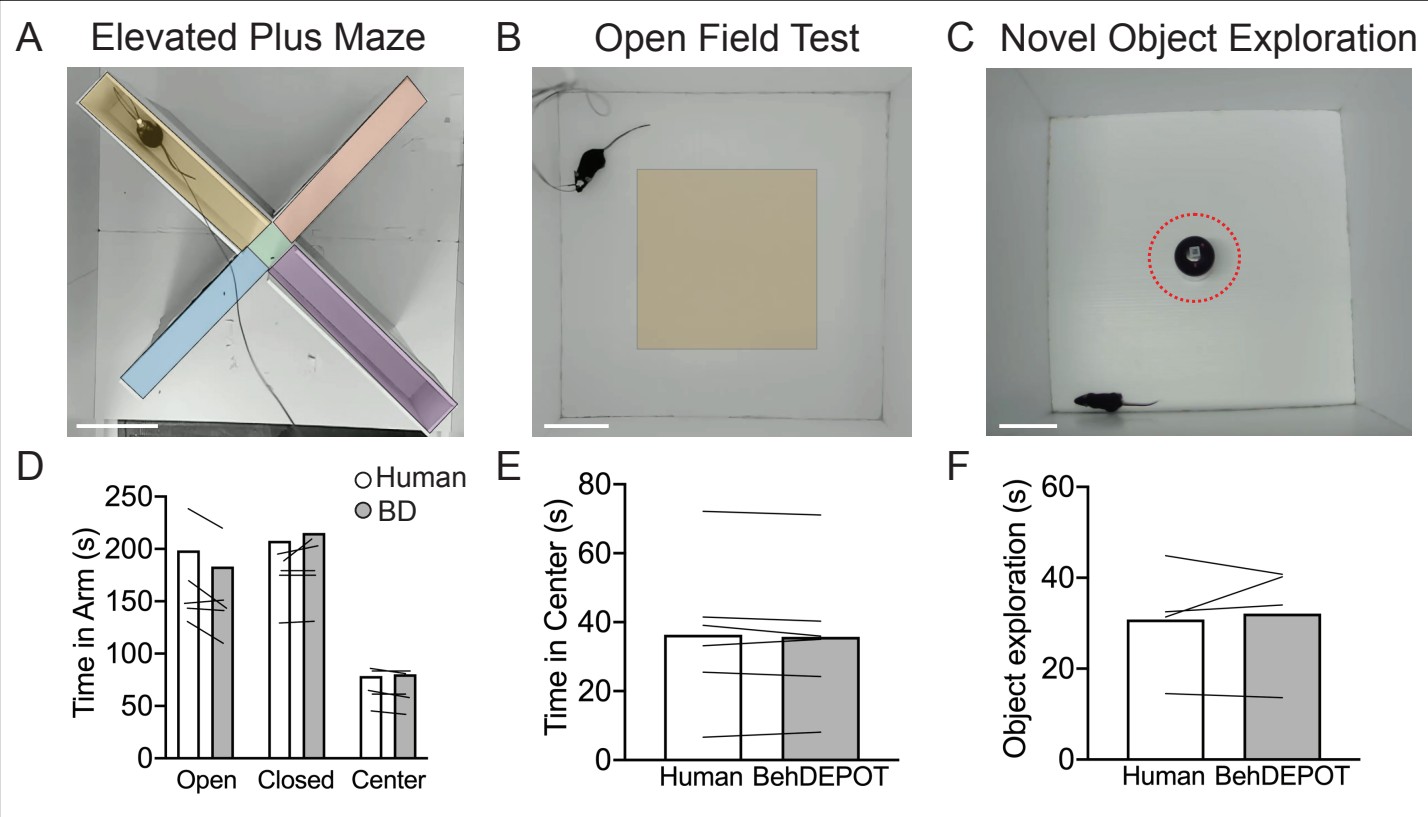

**Figure 5.** Use cases 3–5: EPM, OFT, NOE. (**A–C**) Screens shot from Analysis Module showing user-defined ROIs in the EPM, OFT and NOE. Scale bars, 10 cm. (**D**) Statistical comparison of human vs. BehaviorDEPOT ratings for time in each arm ($F_{Rater}$ (1, 4)=2.260, p=0.21, $F_{Arm}$(2,8)=12.69, P=0.003, Two-way ANOVA; Human vs BD: $P_{Open}$ = 0.15, $P_{Closed}$ = 0.66, $P_{Center}$ = 0.99, Sidak post-hoc test, N=5 mice). (**E**) Statistical comparison of human vs. BehaviorDEPOT ratings for time in center (P=0.50, paired t-test, N=6 mice). (**F**) Statistical comparison of human vs. BehaviorDEPOT ratings for time in center (P=0.66, paired t-test, N=4 mice).

location and specific movements (e.g. head turns in a choice zone, described below). Such assays, including elevated plus maze (EPM), open field test (OFT), novel object exploration (NOE), object location memory, social preference/memory, decision-making assays (T-maze), and working memory assays (Y-mazes), examinenumerous cognitive and emotional processes including anxiety, memory, and decision-making and are published in tens of thousands of studies each year. To extend the utility of BehaviorDEPOT to this broad user base, the Analysis Module includes functions that quantify time spent in user defined ROIs. We demonstrate the utility of these functions in four representative assays: EPM, OFT, NOE, and a T-maze. However, the same functions could be used to analyze numerous assays including Y-mazes, real-time or conditioned place preference, as well as social preference and social memory tests.

## Elevated plus maze, open field test, and novel objection exploration

EPM and OFT are used to measure anxiety-related behaviors in the laboratory (*La-Vu et al., 2020*), but these assays are often scored manually or with expensive software such as Ethovision (*Noldus et al., 2001*). In the EPM, rodents navigate through an elevated (~1 foot off the ground) plus-shaped maze consisting of two open arms without walls and two enclosed arms with walls. Animals that spend more time in the closed arms are interpreted to have higher anxiety (*La-Vu et al., 2020*). In the OFT, rodents are placed in an empty arena without a ceiling for 10–15 minutes. Experimenters measure the fraction of time spent in the perimeter of the box vs. the center. Increased time in the perimeter zone is interpreted as higher anxiety. OFT is also commonly used to measure locomotion in rodents. We trained a DLC network to track animals in an EPM or OFT. We used the BehaviorDEPOT functions to indicate the closed arms, open arms, and center zone for EPM (*Figure 5A*) and the center zone for OFT (*Figure 5B*). BehaviorDEPOT uses the midback keypoint to track the animal as it traverses the

zones, reporting time spent in each zone. BehaviorDEPOT annotations were highly consistent with an expert human rater (*Figure 5D and E* ).

NOE assays can be used to assess exploration and novelty preference (*Ennaceur and Delacour, 1988*; *Leger et al., 2013*; *Vogel-Ciernia and Wood, 2014*; *Zeidler et al., 2020*). An object is placed in the center of an open field and investigation time is measured. NOE is typically defined as instances when the animal's head is within 2 cm of the object and oriented towards the object (*Leger et al., 2013*; *Vogel-Ciernia and Wood, 2014*). After importing DLC tracking data, Behavior-DEPOT allows users to draw an ROI at a set radius around the object (*Figure 5C*). The NOE heuristic uses a combination of head angle and distance with respect to the ROI to label 'investigation' frames. BehaviorDEPOT quantified NOE at levels that were highly consistent with expert human raters (*Figure 5F*).

## Automated quantification of decision-making behaviors

Decision-making is a widely studied area of neuroscience. Many labs study contingency-based, cost-benefit decision-making in mazes that involve choice points. For example, effort-based decision-making involves weighing whether it is worth exerting greater effort for a more valuable outcome. A common rodent assay for effort-based decision-making is the barrier T-maze task in which animals choose whether to climb over a barrier for a large reward (high effort, high value reward; HE/HVR) vs. taking an unimpeded path for a smaller reward (low effort, low value reward; LE/LVR) (*Bailey et al., 2016*). Well trained animals typically prefer HE/HVR choices, taking a direct route over the barrier with stereotyped trajectories. However, when reward or effort contingencies are adjusted, animals demonstrate vicarious trial and error (VTE), thought to be a marker of deliberative (rather than habitual) decision-making. During VTE trials, animals may pause, look back-and-forth, or reverse an initial choice (*Redish, 2016*). Several groups have identified neural signatures of VTE in hippocampus, striatum, and prefrontal cortex suggesting that animals may be playing out potential choices (i.e. vicariously) based on an internal model of maze contingencies (*Redish, 2016*). Simple, flexible, and automated analysis tools for detecting VTE and aligning this behavior with neural data would significantly enhance our understanding of deliberative decision-making.

We used BehaviorDEPOT to automatically detect VTE in a barrier T-maze task and to report the ultimate choice animals made during each effort-based decision-making trial. First, we trained a new DLC network to track mice as they navigated the T-maze. In one version of the task, mice decided whether to turn left to collect a small reward or to turn right and climb a wire mesh barrier for a large reward. We then changed the effort contingency by adding a second barrier to equalize effort across choices. We used the BehaviorDEPOT Analysis Module to perform automated analysis of the task. We defined 8 regions of interest (Approach, Choice, Left Effort, Left Reward, Left Food Cup, Right Effort, Right Reward, and Right Food Cup, *Figure 6A*). We then used the stored tracking data to automatically detect trials, which we defined as the first frame when the animal entered the approach zone until the first frame when the animal entered the reward zone, and to report the outcome of each trial (left choice or right choice) (*Figure 6B–D*).

To develop a VTE heuristic, we used the head angle data stored in the BehaviorDEPOT 'Metrics' data structure. For each trial, we analyzed the head angles when the mouse was in the choice zone and used these values to determine the number of head turns per trial. Manually annotated VTE trials tended to have 1 or more head turns, while non-VTE trials tended to have 0 head turns, so we defined VTE trials as having 1 or more head turns in the choice zone (*Figure 6E*). We then used our BehaviorDEPOT VTE heuristic to detect the fraction of trials with VTE in T-maze sessions with 1 vs 2 barriers, finding a significant increase in the occurrence of VTE trials when a second barrier was added. Importantly, the BehaviorDEPOT performance was highly consistent with a human rater (*Figure 6F*).

Together, these analyses showcase the many functions of the Analysis Module and highlight the utility of the repository of postural and kinematic statistics that is automatically generated in BehaviorDEPOT. By calculating and storing information including velocity, angular velocity, head angle, and acceleration in a framewise manner, BehaviorDEPOT allows users to design automated analysis pipelines for a wide range of commonly studied cognitive tasks. Indeed, BehaviorDEPOT can be tailored to meet individual needs. In the following sections, we describe additional modules that help users develop custom heuristics and integrate them into the Analysis Module.

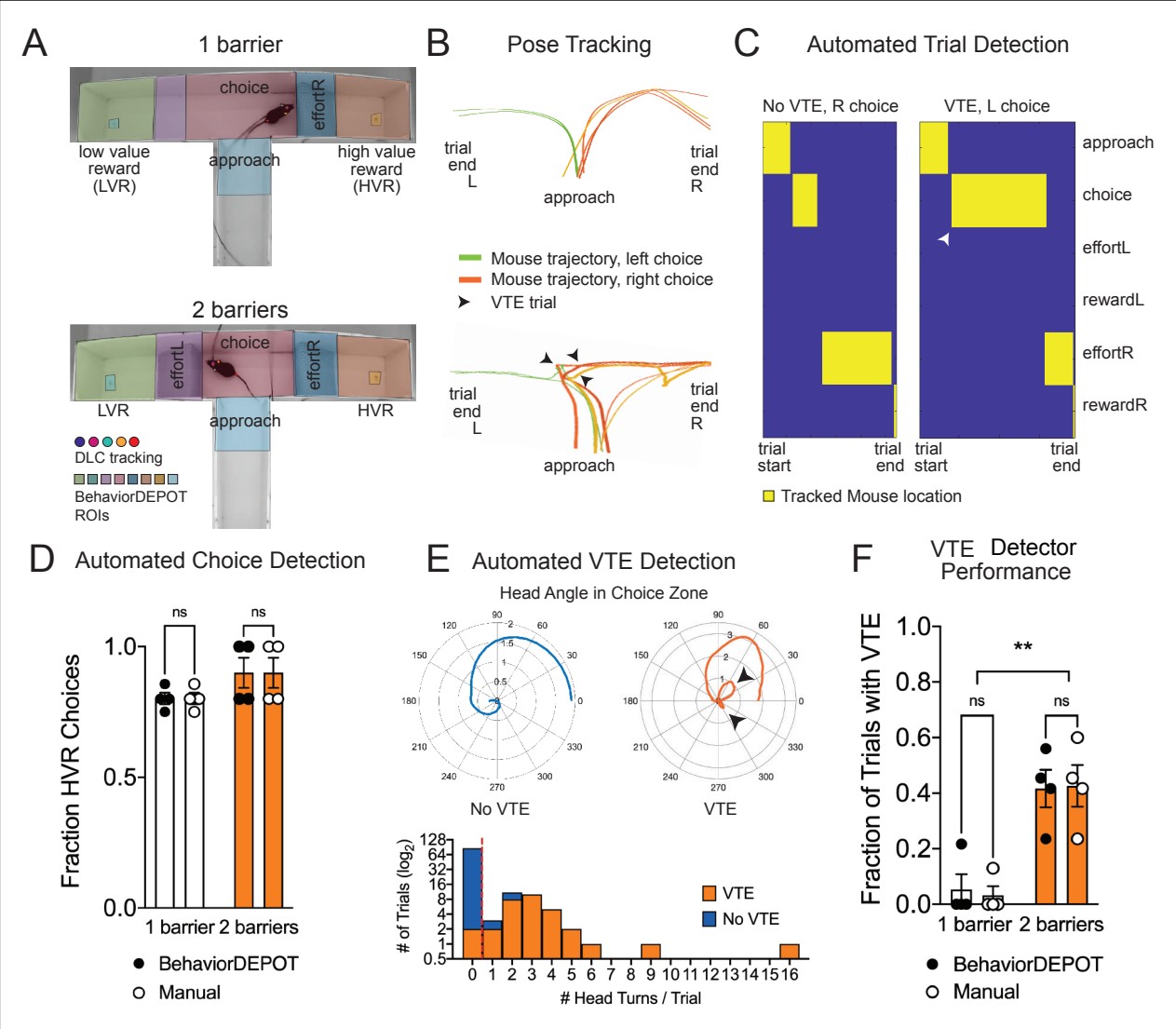

**Figure 6.** Use Case 6: Automated analysis of an effort-based decision-making T-maze task. (**A**) Screen shots showing DLC tracking in a one-barrier (top) and two-barrier (bottom) T-maze and ROIs used for analysis in BehaviorDEPOT. (**B**) Sample mouse trajectories in a one-barrier (top) and two-barrier (bottom) T-maze. Lines represent individual trials for one mouse. Orange lines represent right choices, green lines represent left choices, and thick lines indicate vicarious trial and error (VTE). (**C**) Illustration of automated trial definitions. (**D**) Automated choice detection using BehaviorDEPOT. BehaviorDEPOT indicated choice with 100% accuracy ($F_{Rater}(1,6)$=6.84, $P$>0.99, $F_{Barriers}(1,7)$=4.02, $P$=0.09; $F_{Subject}(6,7)$=0.42, $P$=0.84, two-way ANOVA with Sidak post-hoc comparisons, 84 trials, N=4 mice). (**E**) Top: Polar plots show representative head angle trajectories when the mouse was in the choice zone during a trial without VTE (left) and with VTE (right). Bottom: Histogram of head turns per trial for trials without VTE (blue) and with VTE (orange). Red dotted line indicates selected threshold. (**F**) Fraction of trials with VTE during one-barrier and two-barrier sessions, comparing manual annotations to BehaviorDEPOT classification ($F_{RaterxBarriers}(1,6)$=0.04, $P$=0.85, $F_{Rater}(1,7)$=0.03, $P$=0.85; $F_{Barriers}(1,6)$=22.9, $P$=0.003, two-way ANOVA with Sidak post-hoc comparisons, 102 trials, N=4 mice). Error bars represent S.E.M.

## Developing and optimizing heuristics for behavior detection

To broaden the utility of BehaviorDEPOT, we created four additional modules that guide users through the process of developing their own heuristics. The Inter-Rater, Data Exploration, Optimization and Validation modules help researchers identify and validate the combinations of metrics that track best with their behaviors of interest. Reference annotations can come from human raters or other software.

## The Inter-Rater Module

A major hurdle in developing classifiers or heuristics using supervised approaches is settling on a ground truth definition of the behavior of interest. The Inter-Rater Module compares annotations from

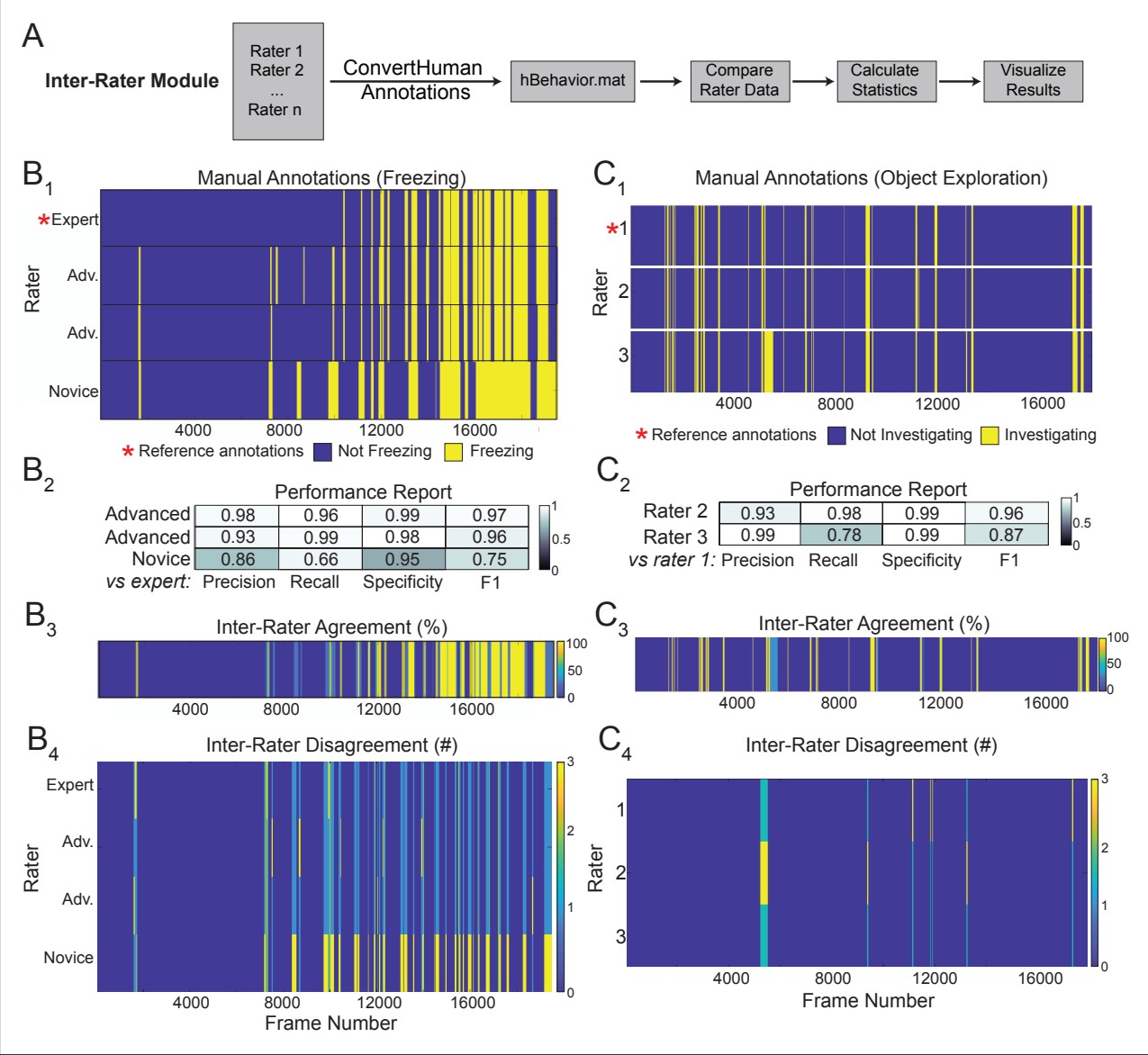

**Figure 7.** Sample outputs of the Inter-Rater Module. (**A**) The Inter-Rater Module imports reference annotations, converts them to a BehaviorDEPOT-friendly format, aligns annotations, and reports statistics about agreement between raters. (**B₁**) Alignment of freezing annotations from four raters with different levels of annotation experience. (**B₂**) Summary report of inter-rater error rates for freezing. (**B₃**) Visualizations of framewise agreement levels for multiple raters for freezing. (**B₄**) Visualizations of framewise disagreements for multiple raters for freezing. (**C₁**) Alignment of NOE annotations from three raters with different levels of annotation experience. (**C₂**) Summary report of inter-rater error rates for NOE. (**C₃**) Visualizations of framewise agreement levels for multiple raters for NOE. (**C₄**) Visualizations of framewise disagreements for multiple raters for NOE.

multiple human raters and identifies points of overlap and disagreement (*Figure 7A*). In response to the initial output of the Inter-Rater Module, human raters can discuss the points of disagreement and modify their manual ratings until they converge maximally. The resulting 'ground truth' definitions of behavior can be used to benchmark the performance of newly developed heuristics or to optimize existing ones.

Here, we demonstrate the outputs of the Inter-Rater module using freezing and novel object exploration as examples. This module imports multiple human annotations and users can select a reference dataset (e.g. the most highly trained expert rater, *Figure 7B1 and C1*). The module compares each set of annotations to the reference, scoring the annotations frame-by-frame as true positive, true negative, false positive, or false negative for each rater. These values are first used to calculate percent overlap and percent error between all raters. Precision, recall, specificity, and F1 score are calculated

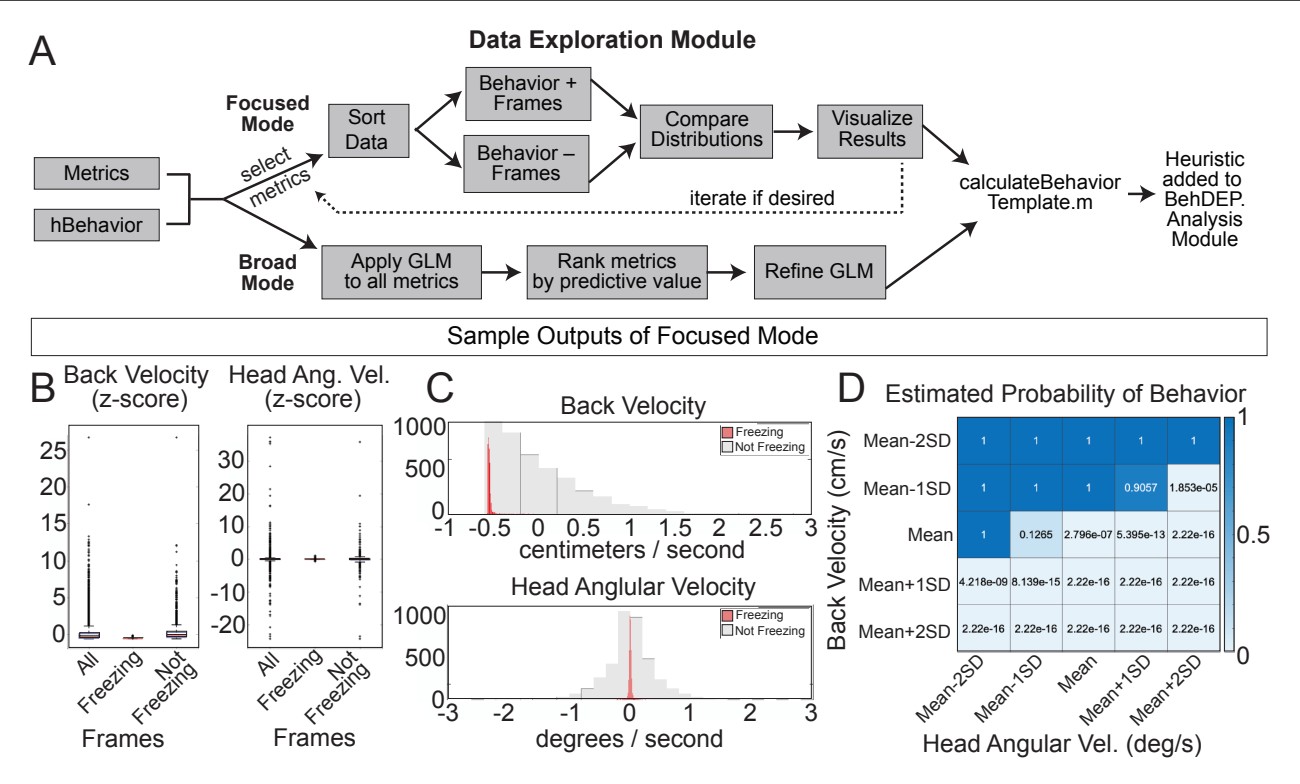

**Figure 8.** The Data Exploration Module. (**A**) The Data Exploration Module takes in metrics from the Analysis Module along with reference annotations. It sorts the data, separating frames containing the behavior of interest from those without and then visualizes and compares the distribution of values for metrics of interest. (**B**) Distributions of Z-scored values for head velocity (left) and change in head angle (right) are distinct for freezing vs. not freezing frames. Box plots represent median, 25th, and 75th percentile. Error bars extend to the most extreme point that is not an outlier (**C**) Histograms showing distribution of Z-scored values for back velocity (top) and head angular velocity (bottom) for freezing (red) vs. not-freezing (grey) frames. (**D**) A generalized linear model (GLM) computes the predictive power of given metrics for frames containing the behavior of interest.

and reported for each rater relative to the chosen reference (*Figure 7B2 and C2*). Additionally, visualizations of frame-by-frame percent agreement (*Figure 7B3 and C3*) and user-by-user disagreement (*Figure 7B4 and C4*) are automatically generated to assist identifying areas of conflict between users. The automated features of the Inter-Rater Module make it fast and easy to perform iterative comparisons of manual annotations, interleaved with human updates, until a satisfactory level of agreement is achieved. The Inter-Rater Module can function as an independent unit and can thus support heuristic development with any application.

## The Data Exploration Module

For users who want to develop new heuristics, the Data Exploration Module helps to identify combinations of keypoint metrics that have high predictive value for behaviors of interest. Users can choose between two different exploration modes: broad or focused. In focused mode, researchers use their intuition about behaviors to select the metrics to examine. The user chooses two metrics at a time and the Data Exploration Module compares values between frames where behavior is present or absent and provides summary data. A generalized linear model (GLM) also estimates the likelihood that the behavior is present in a frame across a range of threshold values for the two selected metrics (*Figure 8A*), allowing users to optimize parameters in combination. Focused mode is fast and easy to use and users can iteratively test as many combinations of metrics as they like.

In broad mode, the module uses all available keypoint metrics to generate a GLM that can predict behavior as well as a rank-order list of metrics based on their predictive weights. Poorly predictive metrics are removed from the model if their weight is sufficiently small. Users also have the option to manually remove individual metrics from the model. Once suitable metrics and thresholds have been identified using either mode, users can plug any number and combination of metrics into a heuristics

template script that we provide and incorporate their new heuristics into the Analysis Module. Detailed instructions for integrating new heuristics are available in our GitHub repository.

We used focused mode to create our freezing heuristic. First, the Data Exploration Module imports the 'Metrics' and 'Behavior' data structures we produced using the Analysis Module along with reference annotations for the same videos. We used the annotations of an expert human rater for reference. In focused mode, users iteratively select combinations of two keypoint metrics (e.g. head velocity, tail angular velocity, etc.) and a behavior label from the human annotations file (e.g. 'Freezing'). We reasoned that low velocity of certain bodyparts would correlate with freezing and thus examined the predictive value of several keypoint velocities. The module creates two data distributions: one containing video frames labeled with the chosen behavior and a second containing the remaining video frames. The larger set is randomly downsampled to ensure that each distribution contains equal numbers of frames and then a series of analyses quantify how reliably chosen metrics align with the behavior of interest. Boxplots (*Figure 8B*) and histograms (*Figure 8C*) identify features that reliably segregate with frames containing a behavior of interest. Indeed, we discovered that the linear velocity of the back and angular velocity of the head tracked particularly well with freezing (*Figure 8B and C*). The GLM revealed the combinations of threshold values that best predicted freezing (*Figure 8D*).

In general, we found that metrics that are well-suited for behavior detection contrast with metrics on frames that do not contain the behavior and have a low standard deviation within the behavior set. Distributions of useful metrics also tend to differ substantially from the total set of frames, especially when compared to frames that do not contain the behavior. The GLM predictions are useful for determining which of the selected metrics best predict the behavior and whether they enhance the predictive value when combined. This module will broaden the utility of BehaviorDEPOT, allowing researchers to tailor its automated functions to fit their needs.

## The Optimization Module

A major advantage of using heuristics to detect behaviors is that parameters can be quickly and easily tuned to optimize behavioral detection in out-of-sample data. Some users of the Optimization Module may have just developed a new heuristic in the Data Exploration Module. Others may want to optimize an existing heuristic for their experimental setup since camera position, framerate and resolution and lighting conditions may influence behavior detection thresholds. To use the module, researchers provide reference annotations and the outputs from the Analysis Module ('Params', 'Behavior', and 'Metrics'). By sweeping through the parameter space for chosen metrics, the Optimization Module identifies the set of thresholds that maximize detection accuracy (*Figure 9A*). Through the graphical interface, users can then update the heuristic threshold values and save the settings for future use. Commercially available freezing classifiers also allow users to adjust the thresholds for freezing detection to ensure that accurate classification can be achieved in a variety of experimental conditions. However, their algorithms are hidden and there is no way to precisely measure error rates. Our Optimization Module adds a level of rigor to the optimization step by reporting heuristic performance as F1 score, precision, recall, and specificity values.

We used the Optimization Module to tune our freezing heuristic for videos recorded in other laboratories. Two research groups provided videos of rats and mice, respectively, recorded using their own acquisition hardware and software. For each dataset, we trained keypoint tracking networks (*Figures 9B1 and 2*) and manually annotated freezing in a subset of the videos. We then analyzed behavior using the Analysis Module with the default settings of our freezing heuristic. Using the 'Parameters', 'Behavior', and 'Metrics' files produced by the Analysis Module, the Optimization Module iteratively swept through a range of threshold values for the four metrics that define the freezing heuristic: back velocity, head angular velocity (*Figures 9C1 and 2*), window width, and count threshold in the convolution algorithm and reported performance for different combinations of values (*Figures 9D1 and 2*).

## The Validation Module

The Validation Module can assess a heuristic's predictive quality by comparing automated behavioral detection to human annotations. The user must first generate a reference set of annotations (either manual or otherwise) and analyze the video using the heuristic of interest in the Analysis Module. In the validation module, the user is prompted to indicate which heuristic to evaluate and to select a

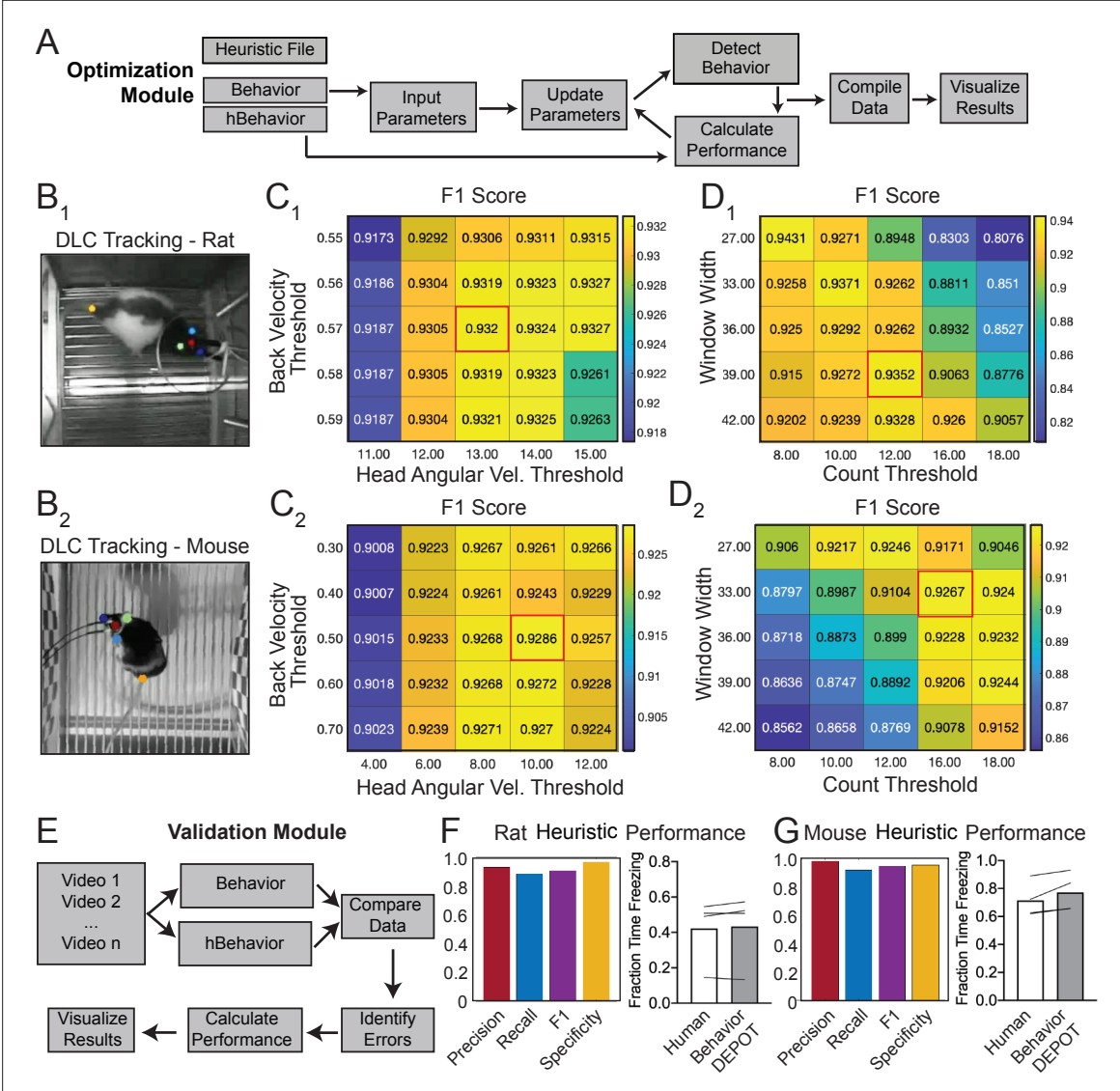

**Figure 9.** Analysis of External Data using Optimization and Validation Modules. (**A**) Optimization Module workflow. This module sweeps through a range of thresholds for metrics calculated based on tracked points and then compares the resulting behavioral classification to human annotations. (**B**$_{1,2}$) DLC tracking in rat and mouse freezing in videos obtained from other laboratories. (**C**$_{1,2}$) Heatmaps showing F1 scores following iterative sweeps through a range of thresholds for two metrics: back velocity and angular velocity of the head. Red box marks highest F1 score. (**D**$_{1,2}$) F1 scores from a subsequent sweep through two additional value ranges for window width and count threshold from the smoothing algorithm. Red box marks highest F1 score. (**E**) Validation Module workflow. (**F**) The BehaviorDEPOT heuristic performed robustly on videos of rats recorded in a different lab (Precision = 0.93; Recall = 0.88; F1=0.91; Specificity = 0.96). BehaviorDEPOT freezing detection was comparable to highly trained human raters (N=4 videos, *P*=0.89, Mann-Whitney U). (**G**) The BehaviorDEPOT heuristic performed robustly on videos of mice recorded in a different lab (Precision = 0.98; Recall = 0.92; F1=0.95; Specificity = 0.95). BehaviorDEPOT freezing detection was comparable to highly trained human raters (N=4 videos, *P*=0.49, Mann-Whitney U).

directory containing behavior videos and the output files from the Analysis Module (Metrics, Behavior, etc). For each video, the module will categorize each frame as true positive, false positive, true negative, or false negative, using the human data as a reference. Precision, recall, specificity, and F1 score are then calculated and visualized for each video. These statistics are also reported for the total video set by concatenating all data and recalculating performance (**Figure 9E**). We validated the performance of the BehaviorDEPOT freezing heuristic on the rat (**Figure 9F**) and mouse (**Figure 9G**) videos acquired in external laboratories which we had optimized using the Optimization Module. In both cases, the optimal combination of thresholds for each video set (**Figure 9C and D**) achieved high F1

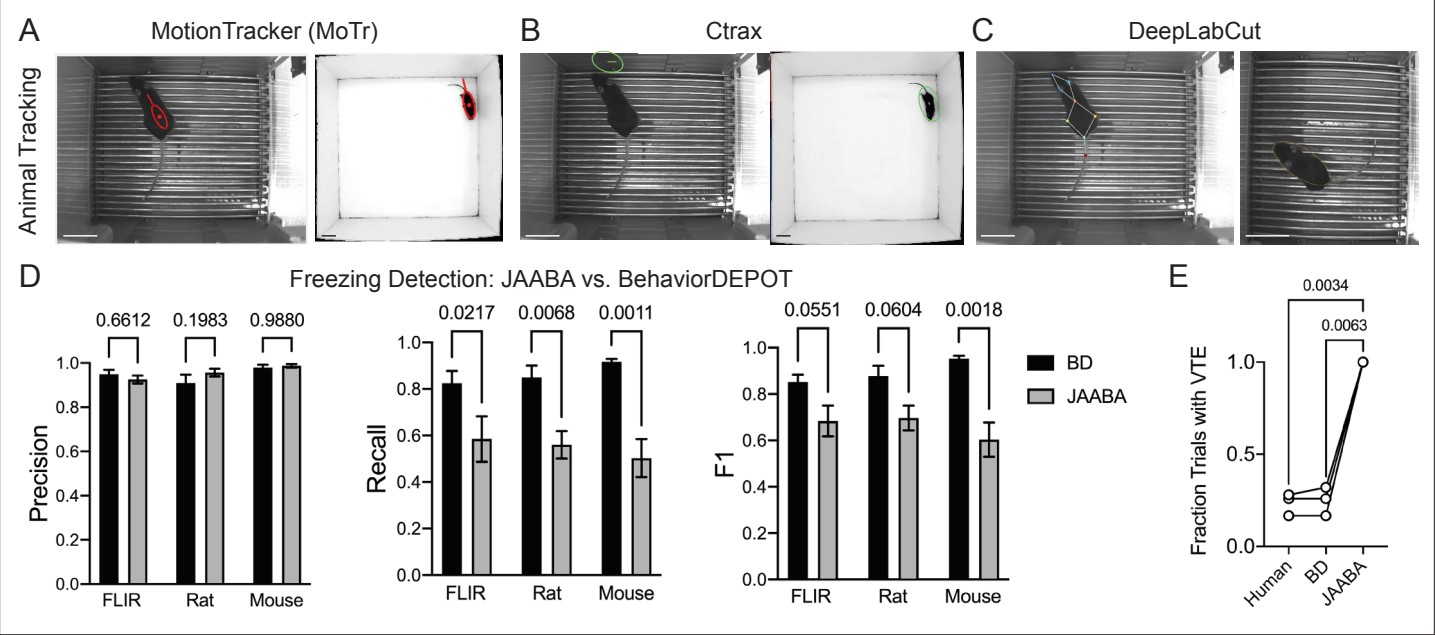

**Figure 10.** Comparisons with JAABA. (**A**) MoTr tracking in a standard fear conditioning chamber (left) and an open field (right). (**B**) Ctrax tracking in a standard fear conditioning chamber (left) and an open field (right). (**C**) DLC tracking in a standard fear conditioning chamber (left) and estimated ellipse based on keypoint tracking (right). (**D**) Quantification of freezing detection errors for BehaviorDEPOT and JAABA (Precision: $F_{Classifier}(1,12)=0.58$, $P=0.46$; Recall: $F_{Classifier}(1,11)=51.27$, $P<0.001$; F1: $F_{Classifier}(1,11)=51.27$, $P<0.001$, N=4–6 videos, 2-way ANOVA with Sidak's multiple comparison test). (**E**) Comparison of VTE detection by human, BehaviorDEPOT, and JAABA ($F_{Classifier}(1.072,2.143)=340.2$, $P=0.0021$, N=3 videos, Repeated measures one-way ANOVA with Tukey multiple comparison's test). Scale bars, 5 cm.

scores (>0.9), indicating that our chosen features for freezing detection are robust across a range of cameras, keypoint tracking networks, experimental conditions, and rodent species.

## Comparison of BehaviorDEPOT and JAABA

Finally, we benchmarked BehaviorDEPOT's performance against JAABA (*Kabra et al., 2013*). Like BehaviorDEPOT, JAABA is geared toward a non-coding audience and its classifiers are based on an intuitive understanding of what defines behaviors. JAABA uses supervised machine learning to detect behaviors and has been widely adopted by the *Drosophila* community with great success. However, fewer studies have used JAABA to study rodents, suggesting there may be some challenges associated with using JAABA for rodent behavioral assays. The rodent studies that used JAABA typically examined social behaviors or gross locomotion, usually in an open field or homecage (*Sangiamo et al., 2020*; *Neunuebel et al., 2015*; *Phillips et al., 2019*; *Nomoto and Lima, 2015*; *van den Boom et al., 2017*).

As the basis for its classifiers, JAABA uses a large feature set that it calculates based on animal poses. JAABA is built around trackers like MoTr (*Ohayon et al., 2013*) and Ctrax (*Branson et al., 2009*), which model animal position and orientation by fitting ellipses. We used MoTr and Ctrax to track animals in fear conditioning videos we had recorded previously. The segmentation algorithms performed poorly when rodents were in fear conditioning chambers which have high-contrast bars on the floor but could fit ellipses to mice in an open field, in which the mouse is small relative to the environment and runs against a clean white background (*Figure 10A–B*). Poor tracking in visually complex environments such as a fear conditioning chamber may explain, at least in part, why relatively few rodent studies have employed JAABA.

To circumvent these issues, we fit ellipses to the mice based on DLC keypoints (*Figure 10C*). Briefly, we used the nose-to-tail distance to calculate the long axis and the hip-to-hip distance to calculate the short axis and used a MATLAB function to generate well-fit ellipses for each video frame. We then imported the ellipse features (animal position (centroid), semi-axis lengths, and orientation) into JAABA. We first trained a freezing classifier using JAABA. Even when we gave JAABA more training data than we used to develop BehaviorDEPOT heuristics (6 videos vs. 3 videos), the BehaviorDEPOT

heuristic for freezing had significantly higher recall and F1 scores when tested on a separate set of videos (*Figure 10D*). We also trained a VTE classifier with JAABA. When we tested its performance on a separate set of videos, JAABA could not distinguish VTE vs. non-VTE trials. It labeled every trial as containing VTE (*Figure 10E*), suggesting a well-fit ellipse is not sufficient to detect these fine angular head movements.

Another strength of BehaviorDEPOT relative to JAABA is that BehaviorDEPOT can quantify behaviors with reference to spatial or temporal cues of interest. JAABA does allow users to draw spatial ROIs but uses this information as the basis of behavioral classification in ways that cannot be detected or controlled by the user. These direct comparisons highlight the need for new behavioral analysis software targeted towards a noncoding audience, particularly in the rodent domain. BehaviorDEPOT can now fill this role.

## Discussion

Keypoint tracking algorithms and in vivo methods including optogenetics, fiber photometry, and miniscope recordings have become increasingly accessible and widely adopted. Because many labs with limited programming expertise want to employ these methods, we need user-friendly software programs that can automate analysis of naturalistic behaviors and facilitate precise alignment of neural and behavioral data.

BehaviorDEPOT is a general utility behavioral analysis software package based on keypoint tracking. Its graphical interface allows even inexperienced coders to run experiments, automatically detect behaviors, and analyze the results of popular assays including those run in operant chambers (e.g. fear conditioning) or spatial mazes (e.g. T-mazes or elevated plus mazes). Our built-in heuristic freezing detector has low error rates across a range of video recording setups, including videos of both mice and rats wearing tethered headmounts. BehaviorDEPOT's flexible interface accommodates varied experimental designs including conditioned tones, optogenetics, and spatial ROIs. It also builds rich data structures that facilitate alignment of neural and behavioral data, and we provide additional code in our GitHub repository for this purpose. The experiment module and the UCLA MiniCAM extend the utility of BehaviorDEPOT, together forming a fully open-source pipeline from data collection to analysis. While the Analysis Module supports many behavioral analyses 'out of the box', we created four independent modules so that researchers can create, test, and optimize new behavioral detection heuristics and then integrate them into the Analysis Module.

BehaviorDEPOT employs heuristics to measure human-defined behaviors based on keypoint tracking data. Keypoints can be used to simultaneously track location and classify fine-scale behaviors. As such, keypoint tracking is well suited for analysis of widely-used assays in which researchers have carefully defined behaviors based on the pose of the animal and its location in an arena. Assays including fear conditioning, avoidance, T-mazes, and elevated plus mazes are published in tens of thousands of studies each year. Yet most still rely on laborious manual annotation, expensive commercial software packages, or open-source software packages that can fail to detect fine movements and are error prone, especially in visually complex arenas. With BehaviorDEPOT, we were able to quickly and intuitively design heuristics that combine spatial tracking with detection of fine movements.

In contrast to methods that use supervised machine learning (*Nilsson et al., 2020*; *Segalin et al., 2021*; *Hong et al., 2015*; *Bohnslav et al., 2021*), BehaviorDEPOT's heuristics are easier to interpret and can be easily tweaked to fit out-of-sample videos. Like any supervised method, BehaviorDEPOT heuristics are based on human definitions, so human bias is not completely absent from the analyses. However, the automation serves to standardize human-expert definitions and can enhance rigor and reproducibility. Moreover, when behaviors are clearly defined, developing reliable BehaviorDEPOT heuristics may require less human-labeled data compared to a supervised machine learning approach. Other methods use unsupervised approaches to classify behavior based on keypoint tracking. While relatively free of human biases, programs like B-SOiD (*Hsu and Yttri, 2021*) and MotionMapper (*Mathis, 2022*) are geared toward analysis of even more subtle behaviors that are more challenging for humans to reliably label and may be out of reach technically for researchers that lack computational expertise. The graphical interface and intuitive design of BehaviorDEPOT heuristics ensure it will serve a broad audience.

BehaviorDEPOT does require users to first train keypoint tracking models. However, the accuracy, precision, and flexibility achieved by these methods will make BehaviorDEPOT more broadly useful

than classifiers based on more coarse detection methods (e.g. background subtraction) that are not robust to environmental complexity or subject appearance. Some freezing detectors use pixel change functions to detect periods of no movement and can operate in complex arenas like fear conditioning chambers. However, they often fail to detect freezing when animals are wearing head-mounts for optogenetics or neural recordings because tethered patch cords can move even when the animal is immobile. To circumvent this problem, ezTrack (*Pennington et al., 2019*), VideoFreeze (*Anagnostaras et al., 2010*), and ANYmaze allow researchers to crop the tether out of the video, but this requires side-view videos and thereby constrains concurrent spatial analyses. In contrast, keypoint tracking algorithms can be trained to detect animals in any experimental condition, so users can analyze videos they have already recorded. With keypoint tracking, researchers can track body parts that best fit their experimental design and can detect fine movements of individual body parts, allowing researchers to detect a wider array of behaviors and discover subtle phenotypes arising from manipulations.

To enhance the utility of BehaviorDEPOT, the Inter-Rater, Data Exploration, Optimization, and Validation Modules help users develop new heuristics or optimize existing ones for their experimental setups. An important step in supervised behavior classification is establishing a ground truth definition of behavior. This is typically achieved by training multiple human raters and through discussion, refining their annotations until they converge maximally. These reference annotations are then used to create classifiers. MARS includes a large dataset of human annotations for social behaviors and provides detailed quantitative descriptions of similarity and divergence between human raters (*Segalin et al., 2021*). While the MARS BENTO feature can import human annotations for visualization alongside behavior videos or automated annotations, our Inter-Rater Module has added functions to automatically calculate performance statistics with the click of a button. This makes it fast and easy to train novice manual raters or establish new ground truth definitions. Human annotations or ratings from another classifier program can be imported into the Data Exploration, Optimization, and Validation Modules and are compared to BehaviorDEPOT metrics or heuristic outputs for development and evaluation of heuristic performance.

Our heuristic for freezing detection operates on similar principles to commercial software that apply a motion threshold and a minimum duration threshold to detect bouts of freezing. However, BehaviorDEPOT has added functionality in that freezing detection can be integrated with spatial analyses so researchers can determine not only when an animal is freezing, but also where it is freezing. Another major limitation of commercial algorithms is that they are not well validated (*Anagnostaras et al., 2000*). Researchers typically use their intuition to set thresholds for freezing. BehaviorDEPOT's Optimization Module allows users to adjust parameters and determine the combination of thresholds that produce the lowest error rates. We envision that new users will manually annotate a set of videos from their lab, analyze them with BehaviorDEPOT using the default heuristic settings, and then test the error rates using the Validation Module. If error rates are high, they can adjust threshold values with the Optimization Module. This process can be done for any behavior of interest. By making the optimization and validation process easy – it can be done with a few button clicks in the graphical interface – BehaviorDEPOT will enhance the rigor of behavior classification and increase reproducibility across labs.

# Materials and methods
## Animals
Female and male C57Bl/6 J mice (JAX Stock No. 000664) aged 10–16 weeks were group housed (2–5 per cage) and kept on a 12 hr light cycle. Following behavior conditioning, animals were individually housed until the memory retrieval sessions. All animal procedures followed animal care guidelines approved by the University of California, Los Angeles Chancellor's Animal Research Committee.

## Contextual fear conditioning
Mice were handled for 5 days preceding the behavioral testing procedure. The conditioning chamber consisted of an 18cm x 18cm x 30 cm cage with a grid floor wired to a scrambled shock generator (Lafayette Instruments) surrounded by a custom-built acoustic chamber. The chamber was scented with 50% Windex. Mice were placed in the chamber and then after a 2-min baseline period, received five 0.75mA footshocks spaced 1 min apart. Mice were removed 1 min after the last shock. Non-shocked

control animals freely explored the conditioning chamber but never received any shocks. The following day, mice were returned to the conditioning chamber and a novel context (different lighting and metal floor, scented with 1% acetic acid), separated by a 1 hr interval. Context presentation order on day 2 was counterbalanced across mice.

## Platform-mediated avoidance

PMA used the fear conditioning chamber described above, except 25% of the floor was covered with a thin acrylic platform (3.5x4 × 0.5 inches). During training, mice were presented with three baseline 30 s 4 kHz tones (CS), followed by nine presentations of the CS that co-terminated with a 2 s footshock (0.13mA). The following day, mice were presented with six CS in the absence of shocks.

## Viruses

AAV1-syn-jGCaMP7f.WPRE (ItemID: 104488-AAV1) was purchased from Addgene and diluted to a working titer of $8.5 \times 10^{12}$ GC/ml. AAV1-CamKIIa-stGtACR2-FusionRed (ItemID: 105669-AAV1) was purchased from Addgene and diluted to a working titer of $9.5 \times 10^{11}$ GC/ml.

## AAV injection with optogenetic cannula implant

Adult wildtype C57/Bl6 mice were anesthetized with isoflurane and secured to a stereotaxic frame (Kopf, 963). Mice were placed on a heating blanket and artificial tears kept their eyes moist throughout the surgery. After exposing the skull, we drilled a burr hole above mPFC in both hemispheres (AP +1.8, ML +/-0.3 from bregma). A Hamilton syringe containing AAV1-CaMKIIa-stGtACR2-WPRE was lowered into the burr hole and 400 nL of AAV was pressure injected into each site (DV –2.25 mm and –2.50 mm from bregma) at 100 nL/min using a microinjector (WPI, UMP3T-1). The syringe was left in place for 10 min to ensure the AAV did not spill out of the target region. After injecting the AAV, chronic fiber-optic cannula (0.37NA, length = 2 mm, diameter = 200 um) were implanted bilaterally above the injection site and secured to the skull with Metabond (Parkell, S371, S396, S398). After recovery, animals were housed in a regular 12 hr light/dark cycle with food and water ad libitum. Carprofen (5 mg/kg) was administered both during surgery and for 2 days after surgery together with amoxicillin (0.25 mg/mL) in the drinking water for 7 days after surgery.

## Optogenetics

Animals were habituated to the patch-cord for 3 days in advance of optogenetic stimulation. A patch-cord was connected to the fiber-optic cannula and animals were allowed to explore a clean cage for 5 min. On the testing day, optical stimulation through the fiber-optic connector was administered by delivering light through a patch-cord connected to a 473 nm laser (SLOC, BL473T8-100FC). Stimulation was delivered continuously with 2.5 mW power at the fiber tip.

## Miniscope surgery and baseplating

For Miniscope recordings, all mice underwent two stereotaxic surgeries (*Cai et al., 2016*). First, adult WT mice were anesthetized with isoflurane and secured to a stereotaxic frame (Kopf, 963). Mice were placed on a heating blanket and artificial tears kept their eyes moist throughout the surgery. After exposing the skull, a burr hole was drilled above PL in the left hemisphere (+1.85, –0.4, –2.3 mm from bregma). A Hamilton syringe containing AAV1-Syn-jGCaMP7f-WPRE was lowered into the burr hole and 400 nL of AAV was pressure injected using a microinjector (WPI, UMP3T-1). The syringe was left in place for 10 min to ensure the AAV did not spill out of the target region and then the skin was sutured. After recovery, animals were housed in a regular 12 hr light/dark cycle with food and water ad libitum. Carprofen (5 mg/kg) was administered both during surgery and for 2 days after surgery together with amoxicillin (0.25 mg/mL) for 7 days after surgery. Two weeks later, mice underwent a GRIN lens implantation surgery. After anesthetizing the animals with isoflurane (1–3%) and securing them to the stereotaxic frame, the cortical tissue above the targeted implant site was carefully aspirated using 27-gauge and 30-gauge blunt needles. Buffered ACSF was constantly applied throughout the aspiration to prevent tissue desiccation. The aspiration ceased after full termination of bleeding, at which point a GRIN lens (1 mm diameter, 4 mm length, Inscopix 1050–002176) was stereotaxically lowered to the targeted implant site (–2.0 mm dorsoventral from skull surface relative to bregma). Cyanoacrylate glue was used to affix the lens to the skull. Then, dental cement sealed and covered the

exposed skull, and Kwik-Sil covered the exposed GRIN lens. Carprofen (5 mg/kg) and dexamethasone (0.2 mg/kg) were administered during surgery and for 7 days after surgery together with amoxicillin (0.25 mg/mL) in the drinking water. Two weeks after implantation, animals were anesthetized again with isoflurane (1–3%) and a Miniscope attached to an aluminum baseplate was placed on top of the GRIN lens. After searching the field of view for in-focus cells, the baseplate was cemented into place, and the Miniscope was detached from the baseplate. A plastic cap was locked into the baseplate to protect the implant from debris.

### Miniscope recordings

Mice were handled and habituated to the weight of the microscope for 4 days before behavioral acquisition. On the recording day, a V4 Miniscope was secured to the baseplate with a set screw and the mice were allowed to acclimate in their home cage for 5 min. Imaging through the Miniscope took place throughout the entire PMA training (~30 min) and retrieval session the following day. Behavior was simultaneously recorded with a UCLA MiniCAM.

### Miniscope data processing and analysis

Frames in which animals were freezing and/or on the safety platform were determined using BehaviorDEPOT. Cell footprints and $Ca^{2+}$ fluorescence timeseries were extracted from Miniscope recordings using MIN1PIPE. We identified 513 neurons across three mice. Custom MATLAB software was used to align data from the behavior camera and the Miniscope camera. To identify neurons that were active during freezing or when the animal was on the safety platform, we plotted receiver operating characteristic curves (ROC) for individual neurons and measured the area under the curve (AUC). ROCs plot the true positive rate (true positive/(true positive +false negative)) against the false positive rate (false positive/(false positive +true negative)) over a range of probability thresholds. Neurons with high AUC values therefore predict the behavioral variable of interest with a high true positive rate and low false positive rates over a large range of thresholds. To determine if a neuron significantly encoded a particular behavioral event, we generated a null distribution of AUCs by circularly shuffling event timing and recalculating the AUC over 1000 permutations. Neurons were considered significantly activated during a behavior if their AUC was greater than 97.5% of AUCs in the null distributions and significantly suppressed if their AUC value was in the lowest 2.5% of all AUCs in the null distribution.

### Histology

Mice were transcardially perfused with phosphate-buffered saline (PBS) followed by 4% paraformaldehyde (PFA) in PBS. Brains were dissected, post-fixed in 4% PFA for 12–24 hr and placed in 30% sucrose for 24–48 hr. They were then embedded in Optimum Cutting Temperature (OCT, Tissue Tek) and stored at –80 °C until sectioning. 60 µm floating sections were collected into PBS. Sections were washed 3x10 min in PBS and then blocked in 0.3% PBST containing 10% normal donkey serum (Jackson Immunoresearch, 17-000-121) for 2 hr. Sections were then stained with rabbit anti-RFP (Rockland 600-41-379 at 1:2000) in 0.3% PBST containing 3% donkey serum overnight at 4 °C. The following day, sections were washed 3x5 min in PBS and then stained with secondary antibody (Jackson Immunoresearch Cy3 donkey anti-rabbit IgG(H+L) 711-165-152, 1:1000) in 0.3% PBST containing 5% donkey serum for 2 hr at room temperature. Sections were then washed 5 min with PBS, 15 min with PBS +DAPI (Thermofisher Scientific, D1306, 1:4000), and then 5 min with PBS. Sections were mounted on glass slides using FluoroMount-G (ThermoFisher, 00-4958-02) and then imaged at 10 x with a Leica slide scanning microscope (VT1200S).

### Open field test

For the open field test, a 50 cm x 50 cm arena with 38 cm high walls was used. The total area of the arena was 2500 $cm^2$ and the center was determined to be one fourth of this, 625 $cm^2$ or a 25 cm x 25 cm square. Mice were placed into the room 20 min prior to test for habituation. At the beginning of the test, mice were placed in the arena facing the wall and allowed to explore for 10 min. Arena was cleaned with 70% ethanol between animals. The total time spent in the center was determined using the OFT heuristic in BehaviorDEPOT and a 2-min epoch was compared to human annotations.

### Elevated Plus Maze

The EPM consisted of a cross-shaped platform 38 cm off the ground with four arms, two that are enclosed by walls 20 cm in height. Mice were placed into the room 20 min prior to test for habituation.

For the test, mice were placed in the center of the platform (5 cm x 5 cm) facing a closed arm and allowed to explore for 8 min. The EPM was cleaned between animals with 70% ethanol. The number of entries into the open arms and the time spent in the open arms were determined using the EPM heuristic in BehaviorDEPOT and compared to human annotations for six full-length videos.

### Novel object exploration

The NOE used the same arena as OF and mice were placed into the room 20 min before testing took place. The objects used were an empty isoflurane bottle (6.5 cm x 6.5 cm x 10 cm) and an ice pack (6 cm x 6.5 cm x 15 cm). At the beginning of each session, the object, cleaned with 70% ethanol, was placed into the center of the arena. The mouse was placed facing a wall into the arena and allowed to explore for 10 min. The arena and objects were cleaned between animals. The time spent exploring the novel object was defined by a human annotator as when the mouse's head was oriented toward the object with the nose within 2 cm of the object. If the mouse was on top of the object, this was not included. These human annotations were compared to the output of BehaviorDEPOT using the NOE heuristic.

### Effort-based decision-making

For T-maze experiments, mice were food deprived to ~85% of the ad libitum initial weight (3–4 days) and habituated to handling and maze exploration. Reward pellets were chopped Reese's peanut butter chips (~0.01 g each). One arm of the maze was designated as high value reward ('HVR', 3 pellets), the other low value reward ('LVR', 1 pellet). Mice were trained to perform effort-reward decision-making via a sequential process. Once mice learned to choose the HVR arm (>80%), a 10 cm wire-mesh barrier was inserted to block that arm. Training was complete when mice achieved >70% success high effort/ HVR choices on 2 consecutive days. Forced trials were used to encourage sampling of the barrier arm during training. Once mice achieved stable performance, a second barrier was inserted in the LVR arm to equalize effort between choices.

### Behavior video recordings

Behavioral videos were acquired using one of the following three setups:

1. 50fps using a Chameleon3 3.2 megapixel monochrome USB camera fitted with a Sony 1/1.8 sensor (FLIR systems, CM3-U3-31S4M-CS) and a 1/1.8 lens with a 4.0–13 mm variable focal length (Tamron, M118VM413IRCS). We recorded 8-bit videos with a 75% M-JPEG compression.
2. 30fps using a ELP 2.8–12 mm Lens Varifocal Mini Box 1.3 megapixel USB Camera.
3. 50fps using a UCLA MiniCAM 5 megapixel CMOS sensor (MT9P031I12STM-DP, ON Semiconductor)

### Manual annotation of behavior

Two-minute samples of each video recording were manually annotated by 1–3 highly trained individuals for freezing behavior using QuickTime7, FIJI (FFMPEG plugin), or MATLAB. One-minute intervals were chosen from the beginning and end of the video recordings to capture diverse behaviors. In some cases, the entire video was annotated. Freezing was defined as the absence of movement except for respiration. Novel object exploration was defined as when the mouse's head was oriented toward the object with the nose within 2 cm of the object. If the mouse was on top of the object this was not included. Time in zones of the EPM and OFT was defined by when the middle of the back crossed the threshold between zones. VTE was defined as when the animal was in the choice zone and swept its head back and forth across ~180°. Annotations were iteratively refined after discussion between users until they converged maximally and then each heuristic was developed based on the annotations of one human expert.

### Design of behavior detection heuristics development and test sets

To develop heuristics, videos were randomly assigned to heuristic development and test sets. Dividing up the dataset by video rather than by frame ensures that highly correlated temporally adjacent frames are not sorted into training and test sets, which can cause overestimation of accuracy. Since the videos in the test set were separate from those used to develop the heuristics, our validation data reflects the accuracy levels users can expect from BehaviorDEPOT. Heuristic performance was

assessed using precision (true positive frames / true positive frames + false positive frames), recall (true positive frames / true positive frames + false negative frames), F1 score (2*(precision*recall / precision + recall)) and specificity (true negative frames / true negative frames + false positive frames).

### Heuristic development

Heuristic parameters for freezing, NOE, and VTE were chosen based on human definitions of these behaviors and examined in a test dataset of videos. For freezing, we explored linear and angular velocity metrics for various keypoints, finding that angular velocity of the head and linear velocity of a back point tracked best with freezing. Common errors in our heuristics were identified as short sequences of frames at the beginning or end of a behavior bout. This may reflect failures in human detection. Other common errors were sequences of false positive or false negative frames that were shorter than a typical behavior bout. We included the convolution algorithm to correct these short error sequences, finding empirically that window widths that are the length of the smallest bout of 'real' behavior and count thresholds approximately one third of the window width yielded the best results.

### Statistical analyses

Statistical analyses were performed in MATLAB or GraphPad Prism.

### The BehaviorDEPOT Pipeline

#### Overview of functions

The Analysis Module imports video recordings of behavior and accompanying DLC keypoint tracking data and exports four data structures (Tracking, Params, Metrics, Behavior) as well as trajectory maps. If using the spatial functions to analyze EPM, OFT, etc., the Analysis Module also outputs a data summary file that reports time spent in each zone as well as number of entries and exits. The four heuristic development modules import these data structures along with reference annotations for videos of interest and report performance metrics (precision, recall, F1 score, specificity) in graphical and text formats. Reference annotations can come from human raters or another classifier software. A list of the inputs and outputs of each module can be found in *Supplementary file 3*.

#### Computer workstation specs

We trained networks in DLC and analyzed videos using two different custom-built workstations (Intel Core i9-9900K processor (8x3.60 GHz/16MB L3 Cache), 2x16 GB DDR4-3000 RAM, NVIDIA GeForce RTX 2070 SUPER - 8 GB GDDR6; AMD RYZEN 9 3950 x processor (16x3.5 GHz/64MB L3 Cache), 16 GB DDR4 RAM, Gigabyte GeForce RTX 2060 SUPER 8 GB WINDFORCE OC). BehaviorDEPOT can run on any personal computer and does not require a GPU.

#### Installation of BehaviorDEPOT

Detailed instructions on BehaviorDEPOT installation can be found on GitHub: https://github.com/DeNardoLab/BehaviorDEPOT. Briefly, after installing a recent version of MATLAB (2018+), BehaviorDEPOT can be downloaded from GitHub and installed with a single click as either a MATLAB application or as a standalone EXE file. The app can be found in the 'APPS' tab of MATLAB. If users would like to read or modify the underlying code, they can hover their computer mouse over the app and then click the link to the file location in the resulting pop-up window. Updates to the application will be added to the GitHub repository as they come available. We welcome feedback and bug reports on the BehaviorDEPOT GitHub page and encourage users to watch the page to be aware of any new releases.

#### DeepLabCut training and model availability

For pose estimation, we found that dedicated models for a given video recording setup produced the lowest error rates for keypoint estimations. It is easy to train keypoint tracking models and worth the initial time investment. To expedite this process for others, our models are available for download from GitHub via a link embedded in the BehaviorDEPOT Wiki page. Depending on the user's setup, our models may be used 'out of the box' or may require some additional training. For high resolution, high framerate cameras (Chameleon-3 camera or MiniCAM), we trained models to track 8 or 9

keypoints (nose, ears, hips, midback, tailbase, mid-tail, and head implant when present). Our webcam videos and the videos we acquired from outside labs had lower resolution and lower framerates, so we trained models to detect 4 or 5 keypoints that were easiest to detect by eye (nose, ears, tailbase, and head implant when present). We trained models in DeepLabCut for 1–1.65 million iterations on sets of ~400 video frames. In our DLC training sets, frame rates ranged from 30–75fps and resolution ranged from 7-32pixels/cm. DLC training errors ranged from 0.74 to 2.76 pixels and DLC test error rates ranged from 1.44 to 5.93 pixels (*Supplementary file 4*). While we do not know the upper limit for DLC tracking errors that can be tolerated by our freezing detection heuristic, our data indicate that heuristic performance is robust to a large range of camera types and DLC models.

## DLC post-processing

BehaviorDEPOT imports keypoint tracking data in the form of a csv or h5 file containing framewise estimates of X-Y coordinates and confidence level for each estimate. After importing the tracking data, BehaviorDEPOT rejects points that were tracked with low confidence ($p < 0.1$), removes outliers using a hampel transformation, smooths the remaining points using a LOWESS (or other user selected) algorithm, and then fills in missing points using spline interpolation. Users can update the confidence threshold in the GUI and the source code can be found in smoothTracking_custom.m. Raw and smoothed tracking data is stored in the data structure 'Tracking' for future reference.

## Metric calculations

To increase the diversity of metrics that can be used to build heuristics, BehaviorDEPOT also calculates additional keypoints defined as the weighted averages of multiple body parts. The raw and smoothed tracking data for calculated points are included in the 'Tracking' structure generated by Behavior-DEPOT (*Supplementary file 1*). For each keypoint, BehaviorDEPOT calculates the linear (cm/s) and angular (deg/s) velocity, acceleration ($cm/s^2$), and linear (cm) and angular (deg) distance moved since the previous frames. All of these data, along with distance travelled throughout the entire section are stored in the 'Metrics' MATLAB structure (*Supplementary file 2*) that is saved automatically for future reference. Metrics and Tracking are then imported by heuristics and spatial tracking functions.

## Behavior analysis functions

### Velocity-based freezing heuristic

This heuristic imports the Metrics structure and identifies frames in which the linear velocity of the back is below a specified threshold value (default is 0.59 cm/s) and angular velocity of the head is below a specified threshold value (default is 15 deg/s). If it meets these requirements, a frame will be labeled as 'freezing'. To smooth over falsely non-contiguous freezing bouts, a sliding window of a specified width produces a convolved freezing vector in which each value represents the number of freezing frames visible in the window at a given frame. An adjustable count threshold then converts the convolved freezing vector into the final binary freezing vector. Finally, the heuristic rejects freezing bouts that are shorter than a user-defined minimum duration. The default value is 0.9 s, a definition that draws on published literature (*Anagnostaras et al., 2000*) and was tested empirically to give the most cohesiveness with human raters. Freezing data is saved framewise as a binarized vector that is the length of the video (Behavior.Freezing.Vector) and boutwise, indicating the start and stop frames of each freezing bout (Behavior.Freezing.Bouts), the length of each bout (Behavior.Freezing.Length), and the total number of bouts (Behavior.Freezing.Count).

### Jitter-based freezing detection Heuristic

To ensure that BehaviorDEPOT will be able to detect freezing for a wide variety of videos and DLC networks, we generated a second freezing heuristic with a different design. As an alternative to the heuristic based on velocity thresholds, we generated a second freezing heuristic that uses a MATLAB changepoint function to find frames at which the mean of the data changes most significantly and then separates frames into groups that minimize the sum of the residual (squared) error from the local mean. The heuristic imports the velocities of the head, nose, and tail from the Metrics structure. Users set the error term, which is calculated based on the estimated max linear velocity for freezing. This minimum residual threshold is ultimately determined by the DLC tracking error, or the 'jitter' of keypoint estimates.

## Novel object exploration heuristic

The object exploration heuristic was developed to automate quantification of object investigation. Animal pose and behavior criteria corresponding to object investigation were developed from multiple sources (*Ennaceur and Delacour, 1988*; *Leger et al., 2013*; *Vogel-Ciernia and Wood, 2014*; *Zeidler et al., 2020*). Key features of behavioral criteria were replicated in the heuristic: (A) nose within an extended boundary from the object (customizable but tested here with 2 cm radius), (B) head oriented directly toward the object (as determined by a line from between the ears extending out past the nose), (C) an exclusionary criteria for climbing on or over the object, created by establishing a smaller boundary within the original object perimeter (customizable but tested at 2 cm from object edge) that rejects any instance in which the nose, ears, or tailbase is within the boundary. To achieve these features, tracking of the ears, nose, and tailbase is required. The object exploration heuristic was manually fit to two different exploration sessions using two different objects: a rectangular, blue plastic object with a flat top that encouraged climbing (6.5x5.5 x 8.5 cm); and a round, dark, glass bottle with a curved top that discouraged climbing (5.5 cm diameter, 13 cm height). The heuristic was validated using BehaviorDEPOT Inter-Rater and Validation modules, comparing the precision, recall, specificity, and F1 scores between heuristic output to an expert human rater across four new videos that were not used to develop the heuristic.

## EPM analysis function

The Elevated Plus Maze analysis function prompts users to define task-relevant ROIs (open arms, closed arms, and center), then calculates the number of entries, total time spent, and percentage time spent in each ROI. The heuristic outputs boutwise and framewise information about when the subject was in each ROI, as well as a report summarizing the results of the session.

## Open field test analysis function

This function prompts the user to define the arena perimeter and the center region. It then calculates the number of entries, total time spent, and percentage time spent in each ROI along with a summary report.

## T-maze analysis function

To analyze T-maze data, we first analyzed the behavior videos using the Analysis Module, drawing 8 ROIs (approach, choice, effortR, effortL, rewardR, rewardL, foodCupL and foodCupR). The VTE heuristic imports data structures produced by the Analysis Module (Params, Metrics, Behavior) and extracts trials. Trial start is defined as when the midback point enters the approach zone, and trial end is defined as when the midback point enters the reward zone. For each trial, the heuristic determines the choice the animal made based on whether it entered the left reward or the right reward zone.

## VTE heuristic

Instances of vicarious trial and error are characterized by sweeps of the head and longer times in the choice zone. The VTE heuristic identifies sequences of frames in the choice zone when the head angles span 180 degrees. It imports outputs from the Analysis Module (Metrics, Params, and Behavior) and uses the intersection of Metrics.degHeadAngle and the choice zone ROI to detect ranges of head angles when the animal was in the choice zone. If the opposing head angles are detected within the choice zone in a single trial, this is counted as a head turn. Trials in which the animal spends at least 0.6 s in the choice zone and has at least 1 head turn aligned best with human annotations of VTE, so the heuristic uses those values and the minimum criterion for VTE detection. This code is available for download from our GitHub repository.

## Heuristic development modules

### The Inter-Rater Module

This module compares annotations from any number of raters. Human annotations can be generated using any software of the experimenters choosing and organized in a three-column csv file (column 1: bout start, column 2: bout stop, column 3: behavior label). The csv file can be converted to a BehaviorDEPOT-friendly MATLAB table using the helper function 'Convert Human Annotations' which

is accessible via BehaviorDEPOT's graphical interface. The Inter-Rater Module imports the resulting 'hBehavior.mat' files and provides visual summaries of rater agreement and disagreement, and reports precision, recall, F1 score, and specificity in both heatmap and tabular (IR_Results.mat) formats.

### The Data Exploration Module

The Data Exploration Module imports the data structures produced by the Analysis Module (Tracking, Params, Metrics, Behavior) and accompanying references annotations as a 'hBehavior.mat' table, which can be generated using the helper function from the graphical interface. Although this function is called 'convertHumanAnnotations', it can be used to convert boutwise annotations from any source into a MATLAB table that can be imported by BehaviorDEPOT. In focused mode, researchers can use their intuition or established definitions of behavior to select pairs of metrics to explore. The module reports metric values in behavior containing and behavior-lacking frames in the form of histograms and box plots and a text file (Results.txt) containing descriptive statistics for the plots. A GLM model is also produced which reports the predictive value of combinations of threshold levels for the two metrics. Users can iteratively examine as many metrics as they please. Once they have settled on features and threshold values, users can use our template script to incorporate the new heuristic into the analysis module.

The broad exploration mode allows researchers to explore the predictive quality of different metrics in an unbiased way. It incorporates all metrics into a GLM and then rejects those with very low predictive value. The resulting refined GLM can be incorporated into the Analysis Module using our heuristic template file.

### Optimization and validation modules

The optimization and validation modules import the data structures produced by the Analysis Module and reference annotations in 'hBehavior.mat' format. They output heatmaps and results tables reporting precision, recall, F1 score, and specificity values.

### The experiment module

The Fear Conditioning Experiment Designer is a MATLAB-based GUI to design and execute cued fear conditioning experiments. It uses an Arduino interface to control commercially available, transistor-transistor logic (TTL)-based equipment. Stimuli are designed to be either auditory or visual. Auditory stimuli can be pure tones or frequency modulated (FM) sweeps. The FM sweeps can be customized for start frequency, end frequency, and sweep duration. Both pure tones and FM sweeps are generated via sine waves in MATLAB, then played through the computer or attached speaker using the sound function. Visual stimuli are generated by sending a timed TTL pulse corresponding to the light pattern. Pure light turns on the TTL for length of the stimulus; pulsed light turns on and off the TTL at a user specific frequency with a user specified duty cycle.

The user builds an experiment in event blocks. Each event can include any permutation of the conditioned stimuli (CS+ or CS-), the shock, and the laser. If a shock is triggered, it is designed to co-terminate with the conditioned stimulus (CS). If a laser is triggered, it occurs for the duration of the CS. Inter-block intervals are randomly chosen between the minimum and maximum values input by the user.

Once the user sets up the event blocks and launches the experiment, a baseline period prior to starting the events begins (if instructed). Afterward, the first event is implemented. This is handled by cycling through a matrix of the events. Each event combination is given a unique identifier, and each identifier has its own implementation function. Within each function, the event is triggered (sound generation or TTL-adjusting), and the timestamp is logged. Timestamps record the system block to the millisecond. For blocks with multiple events (e.g. laser-TTL plus tone plus shock), these are handled sequentially. The pause function is used to time the period between event-on and event-off times.

A countdown of events is shown on screen as each event is being done and completed. Once all events have completed, a file is saved (as 'NAME_YYYY-MM-DD-HH-MM-SS') containing the complete set of experimental information, including identities and parameters of the stimuli, the events and their associated timestamps, and a log of the events delivered in that session.

## MiniCAM instructions and Installation

Descriptions of fabrication and use of MinCAMs can be found on GitHub; *Aharoni, 2022*.

## Acknowledgements

We thank A Klein and N Gogolla for contributing mouse behavioral videos for analysis with Behavior-DEPOT. We thank A Adhikari and P Schuette for helpful advice in designing a freezing heuristic. We thank J Reichl for assistance with DLC network training. We thank C Klune for helpful suggestions on the manuscript. This work was funded by NIH K01MH116264 (L.A.D.), Whitehall Foundation Research Grant (L.A.D), Klingenstein-Simons Foundation Grant (L.A.D.), NARSAD Young Investigator Award (L.A.D.), NIH T32MH073526 (B.J.), ARCS Pre-doctoral Fellowship (C.J.G.), NSF Graduate Research Fellowship (C.M.G) and NIH K08MH116125 (S.A.W.).

## Additional information

### Funding

| Funder | Grant reference number | Author |
| --- | --- | --- |
| National Institutes of Health | K01MH116264 | Laura A DeNardo |
| National Institutes of Health | K08MH116125 | Scott A Wilke |
| Whitehall Foundation | 3 year Research Grant | Laura A DeNardo |
| Simonsen Foundation | Research Grant | Laura A DeNardo |
| ARCS | Pre-doctoral Fellowship | Christopher J Gabriel |
| National Science Foundation | GRFP | Caitlin M Goodpaster |
| National Institutes of Health | T32MH073526 | Benita Jin |
| Brain Research Foundation | BRFSG-2020-02 | Laura A DeNardo |
| Brain and Behavior Research Foundation | 27937 | Laura A DeNardo |

The funders had no role in study design, data collection and interpretation, or the decision to submit the work for publication.

### Author contributions

Christopher J Gabriel, Conceptualization, Data curation, Software, Formal analysis, Supervision, Funding acquisition, Validation, Investigation, Visualization, Methodology, Writing – original draft, Writing – review and editing; Zachary Zeidler, Conceptualization, Data curation, Software, Formal analysis, Validation, Investigation, Visualization, Methodology, Writing – original draft, Writing – review and editing; Benita Jin, Data curation, Software, Formal analysis, Supervision, Funding acquisition, Validation, Investigation, Visualization, Methodology, Writing – original draft; Changliang Guo, Software, Validation, Visualization, Methodology, Writing – original draft; Caitlin M Goodpaster, Data curation, Formal analysis, Funding acquisition, Validation, Investigation, Writing – original draft; Adrienne Q Kashay, Data curation, Investigation; Anna Wu, Formal analysis, Investigation, Methodology; Molly Delaney, Jovian Cheung, Lauren E DiFazio, Investigation; Melissa J Sharpe, Supervision, Investigation, Writing – review and editing; Daniel Aharoni, Conceptualization, Software, Supervision, Validation, Investigation, Visualization, Methodology, Writing – original draft, Writing – review and editing; Scott A Wilke, Conceptualization, Data curation, Formal analysis, Supervision, Funding acquisition, Validation, Investigation, Methodology, Writing – original draft, Writing – review and editing; Laura A DeNardo, Conceptualization, Resources, Data curation, Formal analysis, Supervision, Funding

acquisition, Validation, Investigation, Visualization, Methodology, Writing – original draft, Project administration, Writing – review and editing

### Author ORCIDs
Christopher J Gabriel ⓘ http://orcid.org/0000-0003-3193-2807
Zachary Zeidler ⓘ http://orcid.org/0000-0001-6539-4360
Benita Jin ⓘ http://orcid.org/0000-0002-2580-8618
Caitlin M Goodpaster ⓘ http://orcid.org/0000-0002-2456-9010
Molly Delaney ⓘ http://orcid.org/0000-0002-4464-7282
Melissa J Sharpe ⓘ http://orcid.org/0000-0002-5375-2076
Daniel Aharoni ⓘ http://orcid.org/0000-0003-4931-8514
Laura A DeNardo ⓘ http://orcid.org/0000-0002-7607-4773

### Ethics
All of the animals were handed according to the approved institutional animal care and use committee protocols of the University of California, Los Angeles. The protocol was approved by the University of California, Los Angeles Chancellor's Animal Research Committee (ARC-2019-012-AM-004).

### Decision letter and Author response
Decision letter https://doi.org/10.7554/eLife.74314.sa1
Author response https://doi.org/10.7554/eLife.74314.sa2

---

## Additional files

### Supplementary files
• Supplementary file 1. List and descriptions of tracked keypoints and keypoints calculated by the Analysis Module.

• Supplementary file 2. List and descriptions of metrics calculated by the Analysis Module.

• Supplementary file 3. List of required inputs and automatically generated outputs for every module.

• Supplementary file 4. Descriptions of error rates in DLC keypoint tracking networks and descriptions of held-out video sets used to test each heuristic.

• Supplementary file 5. Descriptions of DLC models used to assess the relationship between DLC mean error and freezing heuristic performance.

• MDAR checklist

### Data availability
All data generated and analyzed during this study are included in the manuscript and supporting file. All code is publicly available on GitHub.

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
