## [Editor Report]

In this manuscript, the authors introduce a new piece of software, BehaviorDEPOT, that aims to serve as an open-source classifier in service of standard lab-based behavioral assays. The key arguments the authors make are that (1) the open-source code allows for freely available access, (2) the code doesn't require any coding knowledge to build new classifiers, (3) it is generalizable to other behaviors than freezing and other species (although this latter point is not shown), (4) that it uses posture-based tracking that allows for higher resolution than centroid-based methods, and (5) that it is possible to isolate features used in the classifiers. These aims are laudable, the software is indeed relatively easy to use, and it will likely be used widely in the community.

---

## [Decision Letter]

**Decision letter after peer review:**

The reviewers have opted to remain anonymous.

*Reviewer #1 (Recommendations for the authors):*

1) Line 24. The authors claim that part of the goal is to remove "human biases" using automated detection. I would push back on this a little. At the end of the day, humans are still choosing which behaviors to classify – there is no representation learning or other similar feature described here. Thus, the aim isn't to eliminate biases, it's really to make those biases more consistent (precision vs. accuracy).

2) Line 108. I'm not entirely sure what a "machine-learning quality camera" is. I would just call it a higher resolution and frame-rate camera.

*Reviewer #2 (Recommendations for the authors):*

I had the following notes about the paper:

No machine learning model is going to generalize well across all possible settings, and it is important for this paper to make clear from the onset what is required for the system to work. Two example points that come to mind are framerate and camera angle. Freezing is described as being detected across frames, which seems like it would be sensitive to changes in video framerate. Does the classifier need to be manually re-adjusted if a different video framerate is used? Also, are there lower bounds on the framerate or resolution of input videos that can be used? Regarding camera angle, all of the example videos shown in this paper are using a camera positioned directly above the animal. Will BehaviorDEPOT work if the mouse is filmed from the side, from below, or from an angle? While it's fine if a given classifier doesn't work in every condition, I do think the authors need to be more explicit in stating the scope of their system.

I like the idea behind the Data Exploration Module, however I was unsure where the explored metrics were coming from. Are these fully hard-coded, or can the user define their own metrics to look at? If metrics are hard-coded, it would be helpful for the authors to provide a complete list of metrics and their definitions within the manuscript (a few lists of example metrics are given, eg lines 84, 121, 146, and 155, but I don't know if these lists are complete.)

Different labs use different software for manual annotation. What annotation file formats are supported by BehaviorDEPOT's Inter-Rater Module? Can it be run on output of commonly used annotation software, like Noldus Observer?

The Methods section is nicely detailed with respect to experimental information, but it needs more information about the data analyses performed. Some examples:

– The "DeepLabCut Training and Model Availability" section of the Methods could use more detail, such as how many frames were trained for each camera, and precise error rates per body part for each video format used. Also despite the heading, there is no information about model availability in this section.

– What software was used for manual annotation of behavior?

– In Use Case 2 (Ca^2+^ imaging), were the computed ROC curves for imaged neurons generated using BehaviorDEPOT, or was this done post-hoc?

The claim that BehaviorDEPOT is "indistinguishable from a highly trained human" is made in reference to an ANOVA comparison of fraction time freezing according to BehaviorDEPOT vs manual annotations. While the authors can use this analysis to claim that BehaviorDEPOT is consistent with human annotations at the level of the entire trial, calling BehaviorDEPOT "indistinguishable from a human" is a much stronger claim, and needs to either be toned down or backed up by additional comparisons. For example, the authors would have to also address BehaviorDEPOT's frame-wise annotations, eg by showing that BehaviorDEPOT's precision, recall, F1 score, and specificity with respect to manual labels are comparable to that of another human.

*Reviewer #3 (Recommendations for the authors):*

There are several competing themes in this manuscript: is it meant to be an easy-to-use detector for simple behaviors once one has DLC data in hand? Or a simplified way to run a few types of popular behavioral experiments? Or a tool for exploring behavior data and performing statistical analyses? Overall a better demarcation of what the software does and doesn't do would be useful.

64: 'The Analysis Module processes pose tracking data from DLC or similar'.. does this mean the specific format, or that any keypoint data can be used? Would this easily generalize to 3D keypoint data or some other type of feature vector to be classified? The ability to take in different data types would add functionality and encourage users.

Different use cases are explored but they each appear to have different types of outputs and I could not find them in the GitHub. Code installation is easy but the guide could use a simpler demo.

[Editors’ note: further revisions were suggested prior to acceptance, as described below.]

Thank you for resubmitting your work entitled "BehaviorDEPOT is a simple, flexible tool for automated behavioral classification based on markerless pose tracking" for further consideration by *eLife*. Your revised article has been evaluated by Laura Colgin (Senior Editor) and a Reviewing Editor.

The manuscript has been improved, but there are some remaining issues that need to be addressed. We ask you to respond to each of the comments below from the reviewers.

*Reviewer #1:*

I thank the authors for their effort in revising this manuscript- the additional detail makes the approach much easier to follow.

Thinking about the project as a whole, I think the authors' description of their freezing/VTE/etc detectors as "classifiers" is setting up false expectations in the reader, as evidenced by all the reviewers asking questions about the choice of algorithm and comparison to tools like SimBA, JAABA, MARS, B-SOiD, etc. The revised text is much clearer about how these detectors are created, and given this description, I think it may make more sense to refer to them as "heuristics", and not as "classifiers" or "algorithmic classifiers": BehaviorDEPOT is for creating human-crafted functions, whereas "algorithmic" and "classifiers" are terms used exclusively to refer to machine learning models that are fit to training data via optimization methods.

Overall, I think BehaviorDEPOT could be a useful tool. The "no coding experience required" claim is somewhat weakened by the fact that it requires users to first train a DeepLabCut model- this is definitely a task that requires a fair amount of coding experience. One could argue that once the pose model is trained, BehaviorDEPOT can be used by a non-coding user to create any number of behavior classifiers; the only shortcoming I see to that argument is that changes to the experimental setup that degrade pose estimator performance (lighting, frame rate, camera type, camera position, objects in the environment) may require periodic pose model retraining even after the initial setup process.

Other aspects of the authors' response to the review were more troubling, and would still need to be addressed. I outline these below.

1) I appreciate the authors' effort in comparing BehaviorDEPOT to JAABA, including processing all of their data with MoTr and Ctrax. I agree that providing more software tools to the rodent community is a respectable aim, and the authors are correct in noting that the additional data post-processing features of BehaviorDEPOT are not provided by JAABA. The authors' analysis satisfactorily addresses Reviewer 1's question about VTE detection.

However, I feel that the performance comparison between BehaviorDEPOT and JAABA in the text and in Figure 10 is inappropriate and misleading. The authors state that they were only able to train JAABA using either (a) the location of the animal's centroid or (b) fit ellipses produced by MoTr and Ctrax, which were of poor quality (as stated by the authors.) It is obvious that a classifier (JAABA) given just the (x,y) coordinate of a mouse's centroid or a poorly fit ellipse is going to be worse for freezing and VTE detection than a classifier (BehaviorDEPOT) given the full pose of the animal-but this undoubtedly because of the quality of the pose representation (MoTr or Ctrax vs DLC), not the quality of the classifier. Unless you can evaluate the two systems on equal footing, i.e. using the same pose representation, you should not present this comparison.

2) Similarly, the authors' response to Reviewer 1's question 3 is focused on the contrast between keypoint-based tracking to earlier classical computer vision methods, which is not what was asked (and is not relevant here, as neither BehaviorDEPOT nor JAABA is a pose estimation tool.) Rather, the key aspect of JAABA's success, as indicated by Reviewer 1, is its use of an optimized version of the GentleBoost algorithm, which takes in a large set of features and selects the small subset that is best suited to classification. I agree with Reviewer 1 that allowing a classifier to select the most informative features is a strong approach that is tough to beat. The authors do make the potentially valid point that classifiers using small sets of hand-crafted features may require less training data, although I don't think enough is shown for them to concretely state that BehaviorDEPOT is more efficient to set up than an appropriately regularized supervised classifier. (For example, how long does it take to annotate training data for a supervised classifier, vs how long did it take to create the new object-investigation detector from scratch?)

3) One last minor point on the subject of classifiers, I'd like to push back on the author's claim that supervised classification tools like SimBA and MARS are for detecting behaviors that are "difficult to see by eye"-this claim doesn't make any sense given that these tools require human annotations to train in the first place.

4) While supplementary Table 4 is helpful, it doesn't answer the question asked, which is "how good does my pose estimation need to be for the classifier to work well"- to do so, you would need to train pose models with different amounts of training data, to see how classifier performance depends on either training set size or mean DLC error, within a given dataset. This would help users figure out whether adding more DLC training data can be expected to improve the performance of the classifier.

5) In response to Reviewer 2's question about camera position, the authors present a new "jitter"-based freezing classifier. They write in the text "in videos recorded from the side, the velocity-based freezing classifier may be slightly affected by distortions in velocity calculations caused by the angle (e.g. when the mouse is moving towards/away from the camera). On the other hand, the 'jitter classifier' should work equally well with any viewing angle as long as tracked points remain generally unobscured." However, in the response to the reviewer, they write "We did not explicitly test our freezing classifier in videos recorded from the side or other angles, but there is no reason to think it wouldn't work since keypoints can be tracked from any angle." It is simply not reasonable to assume a method designed exclusively in one condition will work in another very different condition, let alone work "equally well." This claim is purely speculative and should be removed.

6) The authors also seem to misinterpret Reviewers 2 and 3 questions about annotation and pose data formats. For example, the authors write "BehaviorDEPOT can process pose tracking data in CSV and H5 file formats. Keypoint data collected by any software is fundamentally compatible with BehDEPOT". DeepLabCut data is saved in CSV or H5 formats, but there is additional file structure that must be known to be able to parse those files to extract the data (eg which variables to read, which columns are x vs y coordinates). A different pose estimation tool like SLEAP might still save pose data in an H5 format, but because it uses different file structure conventions, code written to unpack DLC pose data would return an error when run on SLEAP data. While any software is "fundamentally compatible" with BehaviorDEPOT, software other than DLC would require the creation of a conversion script to make sure keypoint data is in the expected format. (As the authors experienced with APT, this is easier said than done, as keypoint storage formats can change with the software version.) The authors should instead perhaps state whether they commit to providing support for new pose formats when requested by users.

*Reviewer #2:*

The authors have responded to the points from the initial review, including added descriptions of how the classifier works as well as a comparison to JAABA.

While testing the demo for ease of use, I found that it does not simply run as downloaded. For example, the demo functions use variables (like Tracked.raw from the csv file) which do not exist. I could fix this in the code myself but if targeted toward an audience with no coding experience, these examples should be simple to run.

---

## [Author Response]

[Editors’ note: The authors appealed the original decision. What follows is the authors’ response to the first round of review.]

Reviewer #1 (Recommendations for the authors):1) Line 24. The authors claim that part of the goal is to remove "human biases" using automated detection. I would push back on this a little. At the end of the day, humans are still choosing which behaviors to classify – there is no representation learning or other similar feature described here. Thus, the aim isn't to eliminate biases, it's really to make those biases more consistent (precision vs. accuracy).

Good point. We edited this paragraph accordingly (now lines 558–560).

Reviewer #2 (Recommendations for the authors):I had the following notes about the paper:No machine learning model is going to generalize well across all possible settings, and it is important for this paper to make clear from the onset what is required for the system to work. Two example points that come to mind are framerate and camera angle. Freezing is described as being detected across frames, which seems like it would be sensitive to changes in video framerate. Does the classifier need to be manually re-adjusted if a different video framerate is used? Also, are there lower bounds on the framerate or resolution of input videos that can be used? Regarding camera angle, all of the example videos shown in this paper are using a camera positioned directly above the animal. Will BehaviorDEPOT work if the mouse is filmed from the side, from below, or from an angle? While it's fine if a given classifier doesn't work in every condition, I do think the authors need to be more explicit in stating the scope of their system.

Like many currently available freezing classifiers (VideoFreeze, ezTrack, FreezeFrame), BehaviorDEPOT allows users to tweak threshold values to best suit their experimental setup. This ensures the classifier will be widely useful. We show that our freezing classifier works across videos that ranged from 7–32 pixels/cm, 30–75fps, mice and rats wearing different tethered headmounts, and videos recorded using 5 different cameras across 3 labs. We included more detailed information in the main text (e.g. line 232–244) and discussion (lines 574–582) along with a table containing all the video recording parameters used for easy reference (Table 4).

We focused on videos filmed from above because we envision that many BehaviorDEPOT users will be interested in integrating spatial tracking with behavioral classification. In videos recorded from the side, much of the spatial information can be lost. We did not explicitly test our freezing classifier in videos recorded from the side or other angles, but there is no reason to think it wouldn’t work since keypoints can be tracked from any angle. We also created a ‘jitter’ based classifier that uses a changepoint function to identify bouts of freezing and works as well as our velocity-based classifier in our videos (Supp Figure 2), but may accommodate a wider range of camera angles, camera qualities, and keypoint tracking errors. We added this paragraph to the manuscript beginning on line 187:

“Since algorithms do not necessarily generalize to all settings, we developed a second freezing classifier that uses a changepoint function to find frames at which the mean velocity changes most significantly and then separates frames into groups that minimize the sum of the residual error from the local mean. Binarized freezing vectors are then processed using the same convolution algorithm and minimum duration threshold as the velocity-based classifier (Figure 2E). This classifier also has high precision and recall scores, performing similarly to the first (Supp. Figure 2). Users can tune the classifier by adjusting a minimum residual threshold for the changepoint function. We termed this the ‘jitter’ classifier since the minimum residual threshold will be determined by the pixel error (aka ‘jitter) in keypoint position estimates. In other words, the threshold will be determined by how much frame-to-frame ‘jitter’ there is in DLC’s estimate of the keypoint location. Keypoint tracking ‘jitter’ may arise as a function of video resolution, framerate, and number of frames used to train a keypoint tracking model. As such, the ‘jitter’ classifier may accommodate a wider range of video qualities and keypoint tracking models. Also, in videos recorded from the side, the velocity-based freezing classifier may be slightly affected by distortions in velocity calculations caused by the angle (e.g. when the mouse is moving towards/away from the camera). On the other hand, the ‘jitter classifier’ should work equally well with any viewing angle as long as tracked points remain generally unobscured.”

In contrast, the ezTrack and VideoFreeze freezing classifiers work well in videos recorded from the side but may not always perform well on videos recorded at other angles. When animals are wearing tethered patch cables, ezTrack users must crop out the cable from the image so that it does not disrupt the freezing detection algorithm. Since it is not possible to crop out the patch cables in videos recorded from above or other angles, the ezTrack classifier may fail in these scenarios. Moreover, there is no straightforward way to validate the freezing classifiers using these methods. BehaviorDEPOT provides the optimization and validation modules so that new users can easily tune and validate our classifiers to suit their experimental setups. Because of this, the BehaviorDEPOT freezing classifier will likely serve a larger audience and yield more reproducible results. We included a paragraph about this in the discussion (lines 596–608).

I like the idea behind the Data Exploration Module, however I was unsure where the explored metrics were coming from. Are these fully hard-coded, or can the user define their own metrics to look at? If metrics are hard-coded, it would be helpful for the authors to provide a complete list of metrics and their definitions within the manuscript (a few lists of example metrics are given, eg lines 84, 121, 146, and 155, but I don't know if these lists are complete.)

We added a table listing the hard-coded metrics (Table 2) and added instructions on our Wiki page to help incorporate their own metrics (https://github.com/DeNardoLab/BehaviorDEPOT/wiki/Customizing-BehaviorDEPOT).

Different labs use different software for manual annotation. What annotation file formats are supported by BehaviorDEPOT's Inter-Rater Module? Can it be run on output of commonly used annotation software, like Noldus Observer?

BehaviorDEPOT simply requires a list of behavior labels and bout start and stop frames. The ‘ConvertHumanAnnotations’ function in the GUI can convert any such file into a BehaviorDEPOT friendly MATLAB structure. Many programs including Noldus Observer save data in a compatible format (CSV) that can be easily imported into BehaviorDEPOT. Because BehaviorDEPOT can import previously recorded manual annotations, it has an edge on programs like JAABA that cannot import previously establish annotations and instead requires users to perform annotations inside their interface.

The Methods section is nicely detailed with respect to experimental information, but it needs more information about the data analyses performed. Some examples:– The "DeepLabCut Training and Model Availability" section of the Methods could use more detail, such as how many frames were trained for each camera, and precise error rates per body part for each video format used. Also despite the heading, there is no information about model availability in this section.

We included methods (lines 856–871) and a table (Table 4) about DLC network training. Sorry we forgot to say where the models are. They are linked in the ReadMe on GitHub (https://github.com/DeNardoLab/BehaviorDEPOT/wiki/Pretrained-DLC-Models). We have now included this information in the methods (lines 841­–857).

The claim that BehaviorDEPOT is "indistinguishable from a highly trained human" is made in reference to an ANOVA comparison of fraction time freezing according to BehaviorDEPOT vs manual annotations. While the authors can use this analysis to claim that BehaviorDEPOT is consistent with human annotations at the level of the entire trial, calling BehaviorDEPOT "indistinguishable from a human" is a much stronger claim, and needs to either be toned down or backed up by additional comparisons. For example, the authors would have to also address BehaviorDEPOT's frame-wise annotations, eg by showing that BehaviorDEPOT's precision, recall, F1 score, and specificity with respect to manual labels are comparable to that of another human.

Point taken. We adjusted our language (lines 173, 289, 339, 347, 382).

Reviewer #3 (Recommendations for the authors):There are several competing themes in this manuscript: is it meant to be an easy-to-use detector for simple behaviors once one has DLC data in hand? Or a simplified way to run a few types of popular behavioral experiments? Or a tool for exploring behavior data and performing statistical analyses? Overall a better demarcation of what the software does and doesn't do would be useful.

BehaviorDEPOT does more than one thing. We added more details to the Results section to make sure all the functions are clearly laid out. We also created Tables 1–3 describing the inputs and outputs of each module.

A brief summary of BehaviorDEPOT’s functions is as follows:

-It is an easy-to-use detector for simple behaviors once one has DLC data in hand.

-It also automates data analysis by quantifying classified behaviors or animal position with reference to user-defined spatial and temporal cues.

-BehaviorDEPOT generates rich data structures for tracking, kinematics, and discrete behaviors to simplify alignment with neural recording data. We provide additional software for Miniscope analysis using the output of BehaviorDEPOT.

-BehaviorDEPOT can help users develop new classifiers

-To ensure that users of the freezing classifier would have a fully open-source pipeline to take them from data collection to analysis, we included a module that can run fear conditioning experiments

64: 'The Analysis Module processes pose tracking data from DLC or similar'.. does this mean the specific format, or that any keypoint data can be used? Would this easily generalize to 3D keypoint data or some other type of feature vector to be classified? The ability to take in different data types would add functionality and encourage users.

BehaviorDEPOT can processing pose tracking data in CSV and H5 file formats. Keypoint data collected by any software is fundamentally compatible with BehDEPOT as long as they are tracking X-Y-coordinate values and likelihood statistics.

Different use cases are explored but they each appear to have different types of outputs and I could not find them in the GitHub. Code installation is easy but the guide could use a simpler demo.

We created an extensive Wiki page on GitHub with detailed description of each module that include screen shots, more demo data, and an FAQ page (https://github.com/DeNardoLab/BehaviorDEPOT/wiki/).

[Editors’ note: what follows is the authors’ response to the second round of review.]

The manuscript has been improved, but there are some remaining issues that need to be addressed. We ask you to respond to each of the comments below from the reviewers.Reviewer #1:I thank the authors for their effort in revising this manuscript- the additional detail makes the approach much easier to follow.

Thank you for the encouraging words.

Thinking about the project as a whole, I think the authors' description of their freezing/VTE/etc detectors as "classifiers" is setting up false expectations in the reader, as evidenced by all the reviewers asking questions about the choice of algorithm and comparison to tools like SimBA, JAABA, MARS, B-SOiD, etc. The revised text is much clearer about how these detectors are created, and given this description, I think it may make more sense to refer to them as "heuristics", and not as "classifiers" or "algorithmic classifiers": BehaviorDEPOT is for creating human-crafted functions, whereas "algorithmic" and "classifiers" are terms used exclusively to refer to machine learning models that are fit to training data via optimization methods.

Good idea. We changed the terminology as suggested. All changes noted with blue text in the manuscript.

Overall, I think BehaviorDEPOT could be a useful tool. The "no coding experience required" claim is somewhat weakened by the fact that it requires users to first train a DeepLabCut model- this is definitely a task that requires a fair amount of coding experience. One could argue that once the pose model is trained, BehaviorDEPOT can be used by a non-coding user to create any number of behavior classifiers; the only shortcoming I see to that argument is that changes to the experimental setup that degrade pose estimator performance (lighting, frame rate, camera type, camera position, objects in the environment) may require periodic pose model retraining even after the initial setup process.

In our experience DeepLabCut is simple and easy to use for training models. It has a graphical user interface, so no manipulation of the underlying code is required to use it (one reason it is so widely implemented). The underlying code itself is well managed and continuously updated (e.g., June 2022 support for a plugin to increase labeling efficiency). In our lab, undergraduate students with no coding experience routinely use DLC to train reliable keypoint tracking networks. It is true that changes in experimental setup may require retraining networks, but we have not found this to be difficult or to require coding experience.

Other aspects of the authors' response to the review were more troubling, and would still need to be addressed. I outline these below.1) I appreciate the authors' effort in comparing BehaviorDEPOT to JAABA, including processing all of their data with MoTr and Ctrax. I agree that providing more software tools to the rodent community is a respectable aim, and the authors are correct in noting that the additional data post-processing features of BehaviorDEPOT are not provided by JAABA. The authors' analysis satisfactorily addresses Reviewer 1's question about VTE detection.However, I feel that the performance comparison between BehaviorDEPOT and JAABA in the text and in Figure 10 is inappropriate and misleading. The authors state that they were only able to train JAABA using either (a) the location of the animal's centroid or (b) fit ellipses produced by MoTr and Ctrax, which were of poor quality (as stated by the authors.) It is obvious that a classifier (JAABA) given just the (x,y) coordinate of a mouse's centroid or a poorly fit ellipse is going to be worse for freezing and VTE detection than a classifier (BehaviorDEPOT) given the full pose of the animal-but this undoubtedly because of the quality of the pose representation (MoTr or Ctrax vs DLC), not the quality of the classifier. Unless you can evaluate the two systems on equal footing, i.e. using the same pose representation, you should not present this comparison.

We regret that we made an error in the text. The text said ‘centroid’, but we actually fit ellipses calculated from our keypoint tracking data. JAABA is built around tracking algorithms that fit ellipses to animals in video timeseries. In its ‘tracks file’, JAABA expects the animal’s X-Y coordinate, orientation, and major and minor axis lengths to generate its feature list (http://jaaba.sourceforge.net/DataFormatting.html#PrepareJAABADataInputFiles), so we agree that using a centroid would not be a fair comparison. We apologize for the confusion generated by this error. In an updated version of our manuscript we have corrected this typo and provided additional details about this comparison (lines 520–529), including a representative image showing the well-fit ellipse (Figure 10C). Since this is the way in which JAABA is intended to work, it is the best comparison possible for this type of data. Considering that we allowed JAABA additional videos for its machine learning classifier, this seems like a fair comparison against which to benchmark BehaviorDEPOT. This clearly demonstrates that for this specific application, BehaviorDEPOT is superior to JABAA.

2) Similarly, the authors' response to Reviewer 1's question 3 is focused on the contrast between keypoint-based tracking to earlier classical computer vision methods, which is not what was asked (and is not relevant here, as neither BehaviorDEPOT nor JAABA is a pose estimation tool.) Rather, the key aspect of JAABA's success, as indicated by Reviewer 1, is its use of an optimized version of the GentleBoost algorithm, which takes in a large set of features and selects the small subset that is best suited to classification. I agree with Reviewer 1 that allowing a classifier to select the most informative features is a strong approach that is tough to beat. The authors do make the potentially valid point that classifiers using small sets of hand-crafted features may require less training data, although I don't think enough is shown for them to concretely state that BehaviorDEPOT is more efficient to set up than an appropriately regularized supervised classifier. (For example, how long does it take to annotate training data for a supervised classifier, vs how long did it take to create the new object-investigation detector from scratch?)

We completely agree that supervised machine learning is a highly effective method for behavior detection. We do not intend to imply that BehaviorDEPOT’s heuristic method is necessarily more efficient to set up than an appropriately regularized supervised classifier. We have carefully gone over the text to ensure we do not unintentionally suggest this and feel the revised version is clear on this point. For instance, we removed the phrase in the paragraphs spanning lines 555­­–575 that suggested that BehaviorDEPOT is more efficient to setup than a regularized supervised classifier and toned down the point about how much human-labeled data is required for BehaviorDEPOT vs. supervised machine learning. While we do show that BehaviorDEPOT is superior to JABAA for this specific freezing application, that does not mean that our approach is always better than a supervised classifier. Instead, we emphasize that BehaviorDEPOT’s heuristics can accurately detect many widely studied behaviors in a way that is easy to use, understand and interpret. The efficiency of using one method over the other likely depends on the specific application and the user’s skill and level of comfort with machine learning. We think that BehaviorDEPOT will be useful for many labs studying the kinds of behaviors we highlight in our manuscript, but undoubtedly there will continue to be many instances where supervised machine learning is the preferred method.

3) While supplementary Table 4 is helpful, it doesn't answer the question asked, which is "how good does my pose estimation need to be for the classifier to work well"- to do so, you would need to train pose models with different amounts of training data, to see how classifier performance depends on either training set size or mean DLC error, within a given dataset. This would help users figure out whether adding more DLC training data can be expected to improve the performance of the classifier.

We added this analysis. We trained 10 separate DLC networks with mean tracking errors that ranged from 1.52–8.65 pixels. We used each network to estimate poses from the same set of 6 videos. We plotted Precision, Recall, Specificity and F1 scores of the freezing classifier against mean DLC error. A linear regression analysis revealed that precision was the most sensitive to DLC tracking errors. This led to a smaller but significant impact on the F1 score as well. This experiment is now described in the main text (lines 184–189) and the plots are included in Figure 2 – Supplementary Figure 1 and descriptions of each model are included in Supplementary File 5.

4) In response to Reviewer 2's question about camera position, the authors present a new "jitter"-based freezing classifier. They write in the text "in videos recorded from the side, the velocity-based freezing classifier may be slightly affected by distortions in velocity calculations caused by the angle (e.g. when the mouse is moving towards/away from the camera). On the other hand, the 'jitter classifier' should work equally well with any viewing angle as long as tracked points remain generally unobscured." However, in the response to the reviewer, they write "We did not explicitly test our freezing classifier in videos recorded from the side or other angles, but there is no reason to think it wouldn't work since keypoints can be tracked from any angle." It is simply not reasonable to assume a method designed exclusively in one condition will work in another very different condition, let alone work "equally well." This claim is purely speculative and should be removed.

Point taken. We removed the claim.

5) The authors also seem to misinterpret Reviewers 2 and 3 questions about annotation and pose data formats. For example, the authors write "BehaviorDEPOT can process pose tracking data in CSV and H5 file formats. Keypoint data collected by any software is fundamentally compatible with BehDEPOT". DeepLabCut data is saved in CSV or H5 formats, but there is additional file structure that must be known to be able to parse those files to extract the data (eg which variables to read, which columns are x vs y coordinates). A different pose estimation tool like SLEAP might still save pose data in an H5 format, but because it uses different file structure conventions, code written to unpack DLC pose data would return an error when run on SLEAP data. While any software is "fundamentally compatible" with BehaviorDEPOT, software other than DLC would require the creation of a conversion script to make sure keypoint data is in the expected format. (As the authors experienced with APT, this is easier said than done, as keypoint storage formats can change with the software version.) The authors should instead perhaps state whether they commit to providing support for new pose formats when requested by users.

We created a conversion script for SLEAP and added a statement to our Github Wiki page committing to provide support for new pose formats when requested by users (https://github.com/DeNardoLab/BehaviorDEPOT/wiki/Inputs).

Reviewer #2:The authors have responded to the points from the initial review, including added descriptions of how the classifier works as well as a comparison to JAABA.While testing the demo for ease of use, I found that it does not simply run as downloaded. For example, the demo functions use variables (like Tracked.raw from the csv file) which do not exist. I could fix this in the code myself but if targeted toward an audience with no coding experience, these examples should be simple to run.

Apologies for this oversight. We carefully tested all the demo data and corrected the errors.